# Metal3D: a general deep learning framework for accurate metal ion location prediction in proteins

Simon L. Dürr ◉[1], Andrea Levy ◉[1] & Ursula Rothlisberger ◉[1] ✉

Metal ions are essential cofactors for many proteins and play a crucial role in many applications such as enzyme design or design of protein-protein interactions because they are biologically abundant, tether to the protein using strong interactions, and have favorable catalytic properties. Computational design of metalloproteins is however hampered by the complex electronic structure of many biologically relevant metals such as zinc . In this work, we develop two tools - Metal3D (based on 3D convolutional neural networks) and Metal1D (solely based on geometric criteria) to improve the location prediction of zinc ions in protein structures. Comparison with other currently available tools shows that Metal3D is the most accurate zinc ion location predictor to date with predictions within 0.70 ± 0.64 Å of experimental locations. Metal3D outputs a confidence metric for each predicted site and works on proteins with few homologes in the protein data bank. Metal3D predicts a global zinc density that can be used for annotation of computationally predicted structures and a per residue zinc density that can be used in protein design workflows. Currently trained on zinc, the framework of Metal3D is readily extensible to other metals by modifying the training data.

Metalloproteins are ubiquitous in nature and are present in all major enzyme families[1,2]. The metals predominantly found in biological systems are the first and second row alkali and earth alkali metals and the first row transition metals such as zinc and copper. Zinc is the most common transition metal (present in ~10% of deposited structures) and can fulfill both a structural (e.g. in zinc finger proteins) or a catalytic role in up to trinuclear active sites. $Zn^{2+}$ is an excellent Lewis acid and is most often found in tetrahedral, pentavalent, or octahedral coordination. About 10% of all reactions catalyzed by enzymes use zinc as cofactor[3].

Metalloproteins are well studied because metal cofactors are essential for the function of many proteins and loss of this function is an important cause of diseases[4]. Industrial applications for metalloproteins capitalize on the favorable catalytic properties of the metal ion where the protein environment dictates (stereo)-selectivity[5–7]. To crystallize proteins, metal salts are also often added to the crystallization buffer as they can help in the formation of protein crystals overcoming the enthalpic cost of association of protein surfaces. Metal ion binding sites can be used to engineer protein-protein interactions (PPI)[8–10] and the hypothesis has been put forward that one origin of macromolecular complexity is the superficial binding of metal ions in early single domain proteins[10].

While simple metal ion binding sites can be rapidly engineered because initial coordination on a protein surface can for example be achieved by creating an i, i+4 di-histidine site on an alpha-helix[11] or by placing cysteines in spatial proximity[12], the engineering of complex metal ion binding sites e.g. in the protein interior is considerably more difficult[2,9] as such sites are often supported by a network of hydrogen bonds. A complication for computational design of metalloproteins is the unavailability of good (non-bonded) force fields for zinc and other transition metals that accurately reproduce (e.g. tetrahedral) coordination with the correct coordination distances which renders design

[1]Laboratory of Computational Chemistry and Biochemistry, Institute of Chemical Sciences and Engineering, Swiss Federal Institute of Technology (EPFL), Lausanne, Switzerland. ✉ e-mail: ursula.roethlisberger@epfl.ch

using e.g. Rosetta very difficult[2,13]. In fact, the latest parametrization of the Rosetta energy function (ref2015)[14] did not refit the parameters for the metal ions which originally are from CHARMM27 with empirically derived Lazaridis-Karplus solvation terms. To adequately treat metal sites in proteins quantum mechanical treatments such as in hybrid quantum mechanics/molecular mechanics (QM/MM) simulations[15,16] is needed whose computational cost is prohibitive for regular protein design tasks. QM/MM simulations can however be used to verify coordination chemistry for select candidate proteins[17]. On the other hand, neural network potentials have been developed for zinc however those require the experimental zinc location as input[18].

Many tools exist to predict whether a protein contains metals (e.g. ZincFinder[19]), which residues in the protein bind a metal (e.g. IonCom[20], MIB[21,22]) and where the metal is bound (AlphaFill[23], FindsiteMetal[24], BioMetAll[25],MIB[21,22]). The input for these predictors is based on sequence and/or structure information. Sequence-based predictors use pattern recognition to identify the amino acids which might bind a metal[26]. Structure-based methods use homology to known structures (MIB, Findsite-metal, AlphaFill) or distance features (BioMetAll) to infer the location of metals. Some tools like Findsite-metal or ZincFinder employ machine learning based approaches such as support vector machines.

Structure based deep learning approaches have been used in the field of protein research for a variety of applications such as protein structure prediction[27,28], prediction of identity of masked residues[29–31], functional site prediction[32,33], for ranking of docking poses[34,35], prediction of the location of ligands[35–39], and prediction of effects of mutations for stability and disease[4,40]. Current state of the art predictors for metal location are MIB[21,22,41], which combines structural and sequence information in the "Fragment Transformation Method" to search for homologous sites in its database, and BioMetAll[25], a geometrical predictor based on backbone preorganization. Both methods have significant drawbacks: MIB excludes metal sites with less than 2 coordination partners from its analysis and is limited by the availability of templates in its database. We assessed both MIB and MIB2, which significantly extended the database of templates. BioMetAll does not use templates but provides many possible locations for putative binding sites on a regular grid. The individual probes in BioMetAll do not have a confidence metric therefore only allowing to rank sites by the number of probes found, which results in a large uncertainty in the position. Both tools suffer from many false positives. In this work, we present two metal ion location predictors that do not suffer from these drawbacks. For both tools, we train solely on zinc and evaluate performance and selectivity for zinc. The deep learning based Metal3D predictor operates on a voxelized representation of the protein environment and predicts a per residue metal density that can be averaged to get a smooth metal probability density over the whole protein. The distance based predictor Metal1D predicts the location of metals using coordination motifs mined from the protein data bank (PDB) directly predicting coordinates of the putative metal binding site. Metal3D paves the way to perform in silico design of metal ion binding sites without relying on predefined geometrical rules or expensive quantum mechanical calculations.

## Results

A dataset of experimental high resolution crystal structures (2085 structures/252324 voxelized environments) containing zinc sites was used for training of the geometric predictor Metal1D and the deep learning predictor Metal3D (Fig. 1). For training of Metal3D, we used the crystal environment including crystal contacts. For predictions, the biological assembly was used.

### Metal3D

Metal3D takes a protein structure and a set of residues as input, voxelizes the environment around each of the residues and predicts the

per residue metal density. The predicted per residue densities (within a $16 \times 16 \times 16$ Å$^3$ volume) can then be averaged to yield a zinc density for the whole protein. At high probability cutoffs the predicted metal densities are spherical (Fig. 2c), at low probability cutoffs the predicted densities are non-regular (Fig. 2a).

We evaluated the quality of the metal densities generated by the model with the discretized Jaccard similarity (Fig. S1) for all environments in the test set. We noticed that at the edges of the residue-centered output densities often spurious density is predicted wherefore we evaluated the similarity of the test set metal density and the predicted metal probability density taking into account a smaller box with zeroed outer edges. Figure S1 shows that the similarity of the boxes does not depend much on the probability cutoff chosen with higher cutoffs yielding slightly higher discretized Jaccard similarity values (0.02–0.04 difference between $p = 0.5$ and $p = 0.9$). Reducing the size of the analyzed boxes (i.e. trimming of the edges) increases the Jaccard similarity from ≈ 0.64 to 0.88 showing that the metal density in the center of the box is more accurate than the density at the edges.

Metal3D is available as self-contained notebook on Google Colab and on Huggingface Spaces.

### Metal1D

The statistical analysis for the geometric predictor uses the LINK records present in deposited PDB structures. A probability map for all zinc coordination motifs was extracted from all training structures (Fig. 1A). The mean coordination distance in the training set was found to be 2.2 ± 0.2 Å, and the default search radius for the predictions was therefore set to 5.5 Å (Table S1). In total 208 different environments with more than 5 different proteins (at 30% sequence identity) were identified. Metal1D is available as self-contained notebook on Google Colab.

### Comparison of Metal1D, Metal3D, MIB and BioMetAll

Existing metal ion predictors can be subdivided into two categories: binding site predictors and binding location predictors. The former identify only the residues binding the ion, the latter predict the coordinates of the metal ion itself. Both Metal1D and Metal3D can predict the coordinates of putative binding sites. We therefore assessed their performance by comparing to recent binding location predictors with available code/webserver: BioMetAll[25],MIB[21] (no longer available as of July 2022) and MIB2[22]. The main tuning parameter (see Table S5) of MIB/MIB2 is the template similarity $t$, with higher values requiring higher similarity of the templates available for the search in structurally homologous metalloproteins. BioMetAll on the other hand was calibrated on available protein structures and places probes on a regular grid at all sites where the criteria for metal binding are fulfilled. The main adjustable parameter for BioMetAll is the cluster cutoff $c$, which indicates how many probes in reference to the largest cluster a specific cluster has. We used the recommended cutoff of 0.5 requiring all chosen clusters to have at least 50% of the probes of the most populous cluster and used the cluster center to compute distances.

We first investigated the potential of all tools to detect the location of a zinc ion binding site in a binary fashion (zinc site or no zinc site). We defined a correctly identified binding site (true positive, TP) as a prediction within 5 Å of an experimental zinc site. In case a tool predicted no metal within the 5 Å radius, we counted this site as false negative (FN). False positive (FP) predictions, i.e. sites where a metal was placed spuriously, were clustered in a 5 Å radius and counted once per cluster. All tools were assessed against the held out test biological assemblies for Metal3D and Metal1D. When the performance of MIB ($t = 1.25$) and BioMetAll is compared against Metal3D with probability cutoff $p = 0.75$ we find that Metal3D identifies more sites (85) than MIB (78) or BioMetAll (75) with a much lower number of false positives (Fig. 3). MIB predicts 180 false positive sites, MIB2 162 sites, BioMetAll 134 sites whereas Metal3D only predicts 9 false positive sites at the

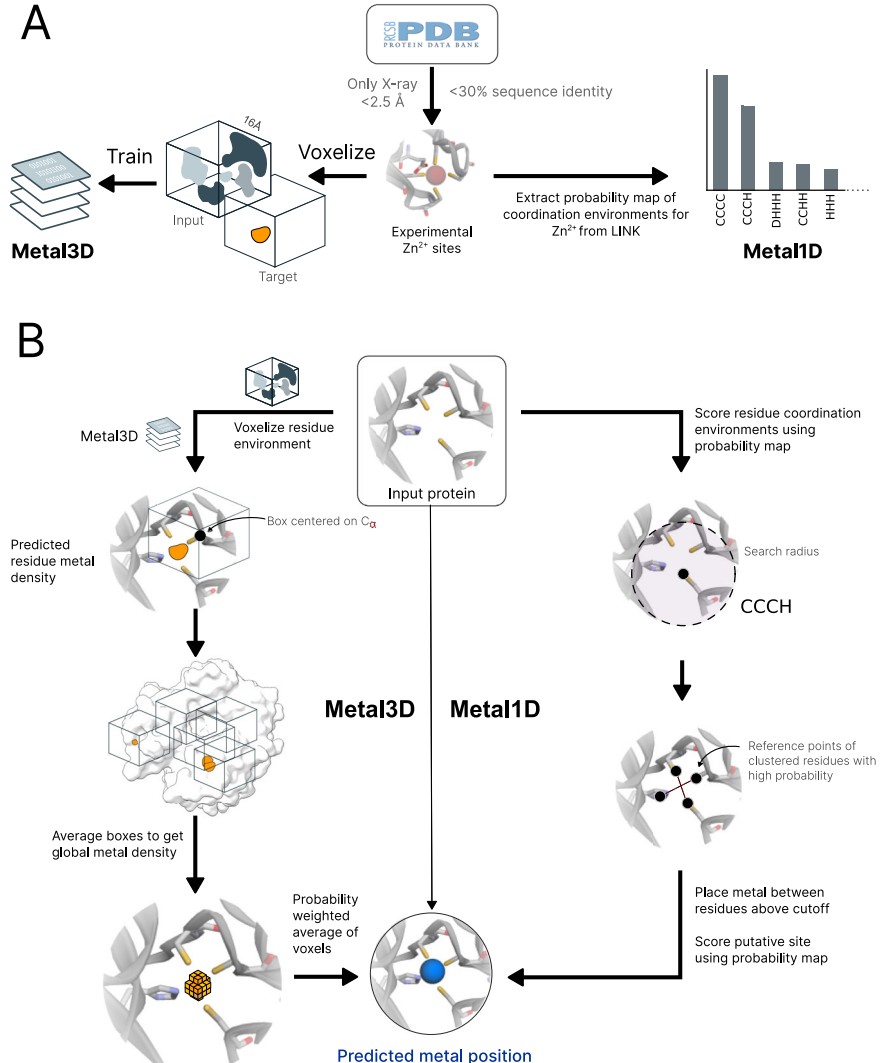

**Fig. 1 | Workflow of Metal3D and Metal1D. a** Training of Metal3D and Metal1D is based on experimental $Zn^{2+}$ sites. Metal1D extracts coordination environments from LINK records, Metal3D is a fully convolutional 3DCNN trained to predict the metal density from voxelized protein environments. **b** In inference mode Metal3D predicts the location of a metal ion by computing per residue metal densities and then averaging them to obtain a global metal density for the input proteins. The

ions can then be placed using the weighted average of voxels above a cutoff. For Metal1D all residues in the protein are scanned for compatibility with the probability map. Metals are placed at the geometric center of residues with high scores according to the probability map. A final ranking of sites is obtained using the probability map.

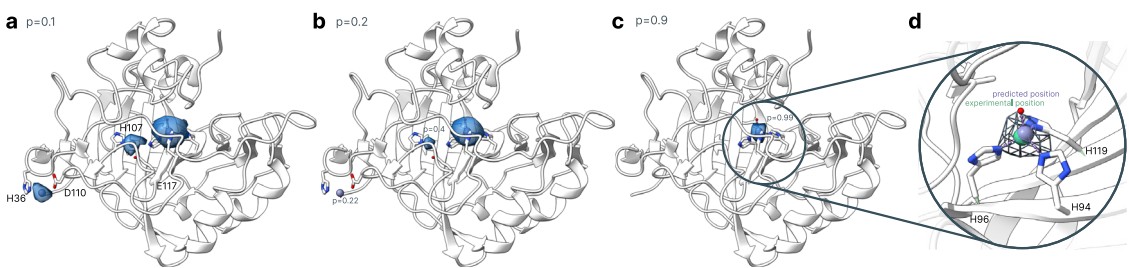

**Fig. 2 | Metal3D probability density.** Probability evolution in HCA2 (PDB 2CBA) for different probability cutoffs. Isosurfaces highlighted at different cutoffs (**a**) $p = 0.1$ (**b**) $p = 0.2$ (**c**) $p = 0.9$ (**d**) inset on active site of HCA2 $p = 0.9$ Predicted positions in darkpurple, experimental position in green, crystal water in red.

$p = 0.75$ cutoff. Metal1D ($t = 0.5$) offers similar detection capabilities (78 sites detected) with a lower number of false positives (47) compared to MIB, MIB2 and BioMetAll. Between MIB and MIB2 the addition of more templates changed the template similarity (Fig. S6). MIB2 has higher recall for low t-scores but reduced precision (Fig. 3B). We removed 70 sites from the list of zinc sites in the test set (189 total) that

had less than 2 unique protein ligands within 2.8 Å of the experimental zinc location and occupancy $< = 0.5$. The amount of correct predictions in this reduced set is almost unchanged for all tools (Fig. 3) indicating that most tools correctly predict sites if they have 2 or more protein ligands. The number of false negatives is reduced for all tools by about 60 sites indicating that most tools do not predict these

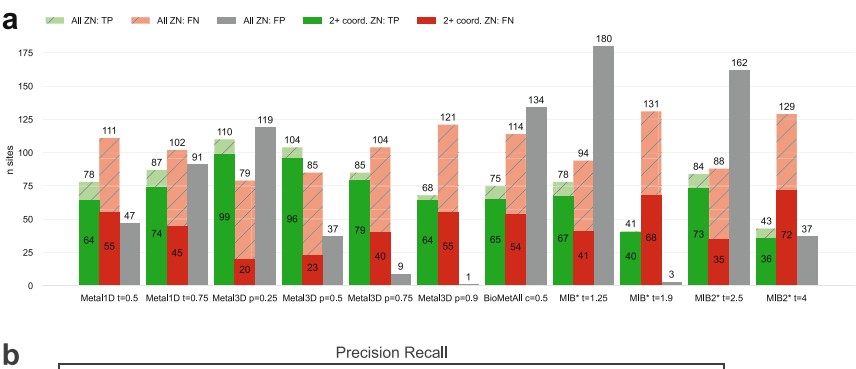

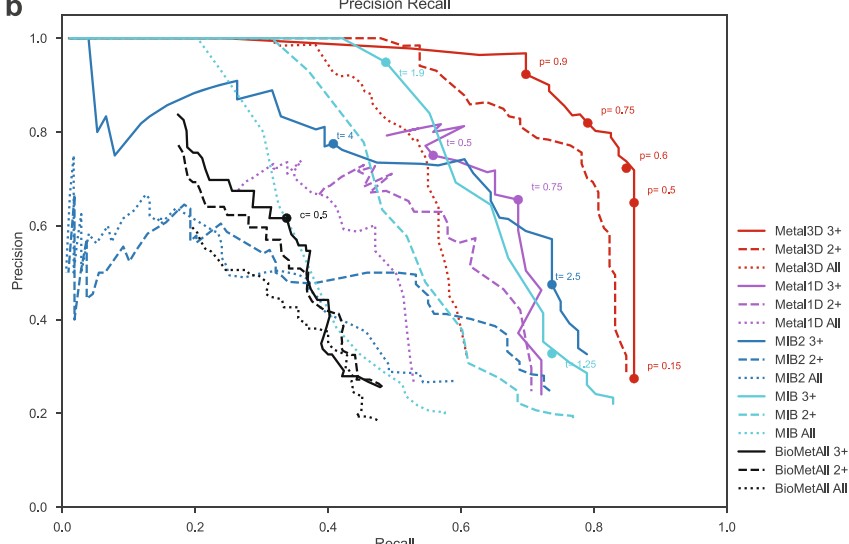

**Fig. 3 | Identification of metal sites within 5 Å. a** Comparison of Metal1D, Metal3D, BioMetAll, MIB, MIB2 on the test set held out from training of Metal1D and Metal3D. Predicted sites are counted as true positives (TP) if they are within 5 Å of a true metal location and as false negatives (FN) otherwise. False positive (FP) probes are clustered and counted once per cluster. The main parameter for each method (t Metal1d, p Metal3D, c BioMetAll, t MIB/MIB2) is explained in Table S5 *For MIB/MIB2 we used 2 structures less because the server did not accept these structures. **b** Precision-Recall curve for all tested tools for the binding site task split into all zinc sites in the test set (dotted lines), sites with 2+ unique coordination partners (dashed lines) and sites with 3+ unique coordination partners (solid lines).

crystallographic artifacts that might depend on additional coordinating residues from an adjacent molecule in the crystal. We also assessed performance on some examples of wrongly modelled metal ions (wrong identity, missing support in electron density) and crystal artefacts contained in the training set showing that Metal3D only predicts ions that have proper support in the structure with high confidence $p > 0.75$ (Fig. S7) and ignores wrongly modelled ions. Of all tools, Metal3D has the least false positives (1 FP at $p = 0.9$) and the highest number of detected sites (110 at $p = 0.25$). The single false positive at $p = 0.9$ does not contain a zinc ion but is a calcium binding site with three aspartates and one backbone carbonyl ligand (Fig. S5). For physiological sites with 3+ unique protein ligands Metal3D probabilities are all above 50% (Fig. 3B).

After assessment of how many sites the tools predict, another crucial metric is the spatial precision of the predictions. For the correctly identified sites (TP) we measured the mean absolute deviation (MAD) between experimental and predicted position (Fig. 4a). The MAD for Metal3D at $p = 0.9$ is $0.70 \pm 0.64$ Å and $0.74 \pm 0.66$ Å at $p = 0.25$ indicating that low confidence predictions are still accurately placed inside the protein. The median MAD of predictions for Metal3D at $p = 0.9$ is 0.52 Å indicating that for half of the predictions the model predicts at or better than the grid resolution of 0.5 Å .

BioMetAll is not very precise with a MAD for correctly identified sites of $2.71 \pm 1.33$ Å . BioMetAll predicts many possible locations per cluster with some of them much closer to the experimental metal binding site than the cluster center. However, it does not provide any

ranking of the probes within a cluster and therefore the cluster center was used for the distance calculation. Metal1D $t = 0.5$ (MAD $2.06 \pm 1.33$ Å) which identifies more sites than BioMetAll is also more precise than BioMetAll. MIB $t = 1.9$ detects sites with high precision (MAD $0.77 \pm 1.09$ Å) but it relies on the existence of homologous sites to align the found sites. MIB2 $t = 2.5$ is less precise (MAD $0.89 \pm 1.00$ Å) than MIB.

## Selectivity for other metals

Both Metal3D and Metal1D were exclusively trained on zinc and we assessed their performance on sodium (Na$^+$, PDB code NA), potassium (K$^+$, PDB code K), calcium (Ca$^{2+}$, PDB code CA), magnesium (Mg$^{2+}$, PDB code MG), and various transition metals (Fe$^{2+}$, Fe$^{3+}$, Co$^{2+}$, Ni$^{2+}$, Cu$^{2+}$, Mn$^{2+}$ with corresponding PDB codes FE2, FE, CO, NI, CU, MN, respectively) from 100 randomly drawn structures from the clustered PDB at 30% identity not used for training. For NI (93), CU (68), FE2 (57) and CO (30) less sites were used. Only sites with at least 3+ unique protein ligands and occupancy > 0.5 were used for the analysis to exclude crystallographic artifacts and use only highly defined sites which should exhibit most selectivity towards a specific metal. Figure 4B shows that recall for Metal3D is high for all transition metals, meaning that the model correctly finds most sites even though it was only trained on zinc. For the alkali and earth alkali metals recall is much lower as the model only finds some sites. The mean probability for found zinc sites (ZN $p = 0.97 \pm 0.05$) in the test set is higher than for the other transition metals (Fig. S3) and

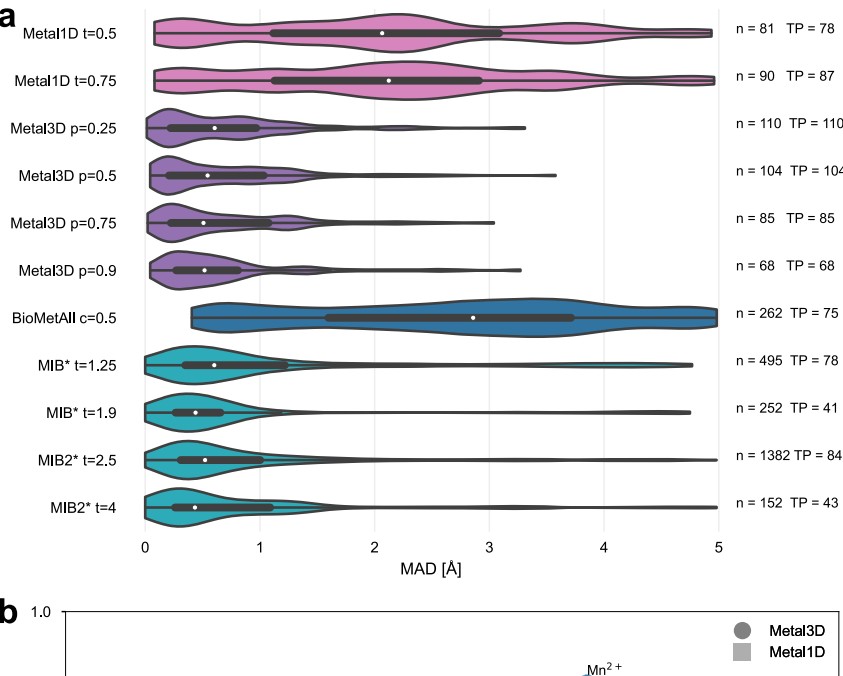

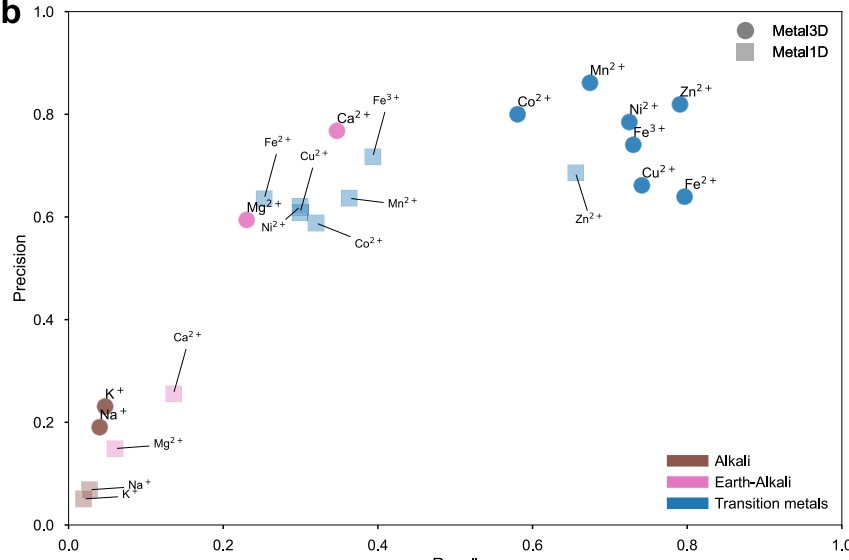

**Fig. 4 | Mean absolute deviation of correctly predicted sites and selectivity for other ions. a** For all sites where predicted and experimental location are available (true positives in Fig. 3) we compute the mean absolute deviation (MAD) using Metal1D, Metal3D, BioMetAll, MIB and MIB2 on the test set. Some tools predict multiple locations (n) per site (TP). *For MIB and MIB2 we used 2 structures less because the server did not accept these structures. For each tool the whisker plot indicates the median (white dot) and the first quartiles (black box), kernel density estimation of all data points shown as violin plot with minima and maxima indicated by whiskers. **b** Precision and recall for the test set and 100 randomly drawn structures (except 93 for NI, 68 for CU, 57 for FE2, CO for 30) for other transition (blue), earth-alkali (pink) and alkali (brown) metals. Metal1D as light squares, Metal3D as circles. Threshold used for Metal3D $p = 0.75$, threshold used for Metal1D $t = 0.75$.

significantly higher than the probability for alkali metals (NA $p = 0.82 \pm 0.06$, K $p = 0.88 \pm 0.06$) while the probability for the earth alkali metals is slightly higher with MG ($p = 0.89 \pm 0.06$) similar to CA ($p = 0.92 \pm 0.07$). The MAD for each found metal site is again lowest for zinc ($0.52 \pm 0.45$ Å). The MAD for the earth-alkali and alkali ions are higher than for the transition metal ions (Fig. 4A).

Structures where a sodium is detected by Metal3D (such as 2OKQ[42], 6KFN[43]) have at least 2 side chain coordinating ligand atoms and only one backbone (2OKQ) or no backbone ligand atom (6KFN). Canonical sodium binding sites e.g. such as in PDB 4I0W[44] with two coordinating backbone carbonyl oxygen atoms and one asparagine side chain have probabilities around 5% and are basically indistinguishable from background noise of the model. For Metal1D overall recall is lower with a clearer distinction of main group versus transition

metals compared to Metal3D. For Metal1D also a larger gap between zinc and other transition metals exists (Fig. 4).

## Multi-nuclear metal centers

We assessed Metal3D for its performance on multi-nuclear metal sites (Fig. S8) and also collected statistics on their prevalence in training and test sets (Figs. S9, S10). For the di-nuclear NDM-1 protein (PDB 4EYL) Metal3D produces a density map that well reproduces both metal ions (coordination motifs His₃, HisAspCys) which are separated by 4.0 Å . There is a third spurious prediction in vicinity to the active site with no experimental support for metal binding in the structure. This site has a realistic coordination motif (HisGluAsp) and $p = 0.74$, which is higher than the probability predicted for the HisAspCys site $p = 0.66$. The clustering which places individual probes in the density map works

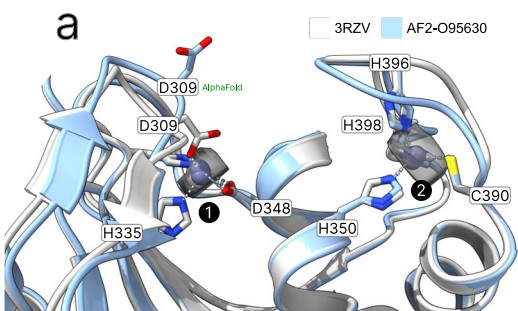

**Fig. 5 | Annotation of AlphaFold2 structures. a** Predicted metal binding sites (**a, b**) from Metal3D, respectively AlphaFill compared to the experimentally found zinc positions for Uniprot O95630. Metal3D places the metal with high accuracy even if sidechains are not perfectly predicted by AlphaFold for site 1. **b** Palmitoyltransferase ZDHHC23 (Uniprot Q8IYP9) and ZDHHC15 (Uniprot Q96MV8). AlphaFill can only place the metal for ZDHHC15 because sequence identity for ZDHHC23 is only 24%. Probability isosurfaces from Metal3D for both structures at $p = 0.6$, colored in gray.

well for the metal that is coordinated by His$_3$ but not for the other which is coordinated by HisAspCys. At higher isovalues for the probability density increased support is present for the His$_3$ motif (max $p = 0.97$) compared to the other motif (max $p = 0.66$). With the default settings (clustering threshold 0.15, distance threshold 7 Å) only one probe is placed. Two probes can be obtained by setting the clustering threshold to 0.5 separating the probability densities for each metal site. Two other examples were extracted from the trainset after literature review[45]: Alkaline phosphatase and Phospholipase C. Both enzymes have tri-nuclear metal centers and were contained in the training set. For Phospholipase C we analyzed the structure contained in the training set (1AH7). Metal3D correctly identifies 2 of 3 metal sites. The metal site with one backbone coordinating amino acid has some density extending to the identified metal site close to it (3.5 Å) but not enough to place a separate zinc there after clustering. For Alkaline phosphatase we analyzed a structure (PDB 1ALK) that was not directly contained in the training set which contained two Zn$^{2+}$ and one Mg$^{2+}$ ion. The structure contained in the training set (PDB 5C66) instead has three Zn$^{2+}$ modelled. Metal3D correctly identifies the two Zn$^{2+}$ modelled in 1ALK but not the magnesium even though it was trained on 5C66 containing a Zn$^{2+}$ in this position. The site has a threonine coordinating the ion, which is not common for zinc.

### Annotation of AlphaFold 2 structures

AlphaFold2 often predicts side chains in metal ion binding sites in the holo conformation[27]. Tools like AlphaFill[23] use structural homology to transplant metals from similar PDB structures to the predicted structure. Metal3D does not require explicit homology based on sequence or structural alignment like AlphaFill so it is potentially suited to annotate the dark proteome that is now accessible from the AlphaFold database with zinc binding sites. Metal3D identifies both the catalytic site (1) and the zinc finger (2) for the example (PDB 3RZV[46], Fig. 5 A) used in ref. 23 with high probability ($p = 0.99$) even though one of the sites in the AlphaFold model is slightly disordered with one of the binding residues in the solvent facing conformation (D309). The distances between predicted and modeled metal locations for Metal3D are 0.22 Å and 0.37 Å, for AlphaFill they are 0.21 Å and 0.41 Å.

AlphaFill uses a 25% sequence identity cutoff which can be problematic for certain proteins with no structurally characterized homologues. For *human* palmitoyltransferase ZDHHC23 (Uniprot Q8IYP9) a high confidence AlphaFold2 prediction exists but AlphaFill cannot place the zinc ions because the sequence identity is 24% to the closest PDB structure (PDB 6BMS[47]), i.e. below the 25% cutoff. For the identical site in another *human* palmitoyltransferase ZDHHC15 (Uniprot Q96MV8) AlphaFill is able to place the metal because of higher sequence identity to 6BMS (64%) (Fig. 5 B). For ZDHHC23 Metal3D is able to place the metal with high confidence (MAD 0.75 Å

for site 1 and 0.48 Å for site 2, $p > 0.99$) based on the single input structure alone.

### Metal3D for metalloprotein engineering

Human carbonic anhydrase II (HCA2) is a well studied metalloenzyme with a rich amount of mutational data available. For the crystal structure of the wildtype enzyme (PDB 2CBA[48,49]), Metal3D recapitulates the location of the active site metal with a distance deviation to the true metal location of 0.21 Å with a probability of $p = 0.99$. At lower probability cutoffs ($p < 0.4$) the probability map indicates further putative metal ion binding sites with interactions mediated by surface residues (e.g. H36, D110, $p = 0.22$) (Fig. 2).

To investigate the capabilities for protein engineering we used mutational data for first and second shell mutants of the active site residues in HCA2 with corresponding $K_d$ values from a colorimetric assay[50]. For most mutants no crystal structures are available so we used the structure builder in the EVOLVE package to choose the most favorable rotamer for each single point mutation based on the EVOLVE-ddg energy function with explicit zinc present (modeled using a dummy atom approach[51]). The analysis was run for every single mutant and the resulting probability maps from Metal3D were analyzed. For the analysis we used the maximum predicted probability as a surrogate to estimate relative changes in $K_d$. For mutants that decrease zinc binding drastically we observe a drop in the maximum probability predicted by Metal3D (Fig. 6). The lowest probability mutants are H119N and H119Q with $p = 0.23$ and 0.38. The mutant with the largest loss in zinc affinity H94A has a zinc binding probability of $p = 0.6$. Conservative changes to the primary coordination motif (e.g. H → C) reduce the predicted probability by 10–30%. For second shell mutants the influence of the mutations is less drastic with only minor changes in the predicted probabilities.

### Discussion

Metal3D predicts the probability distribution of zinc ions in protein crystal structures based on a neuronal network model trained on natural protein environments. The model performs a segmentation task to determine if a specific point in the input space contains a zinc ion or not. Metal3D predicts zinc ion sites with high accuracy making use of high resolution crystal structures (<2.5 Å). The use of high resolution structures is necessary because at resolutions greater than the average zinc ligand coordination distance (2.2 Å) the uncertainty of the zinc location noticeably increases[52] which would likely hamper the accuracy of the site prediction.

In contrast to currently available tools, for Metal3D, it is not necessary to filter the training examples for certain coordination requirements (i.e. only sites with at least 2 protein ligands). The model thus sees the whole diversity of zinc ion sites present in the PDB. Such a

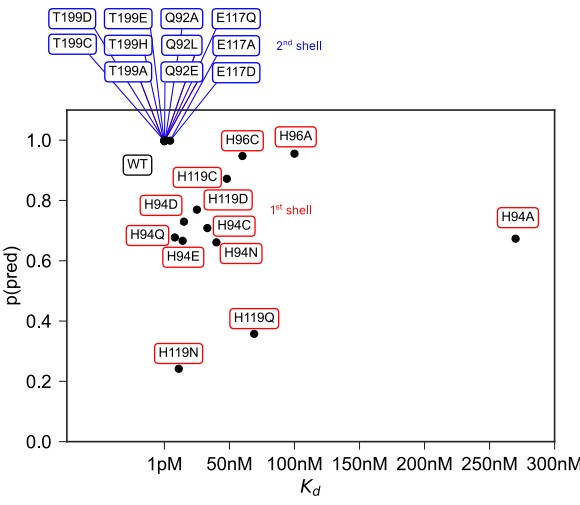

**Fig. 6 | Protein design application.** Experimentally measured $K_d$ values[65-69] for 1st and 2nd shell active site mutants of HCA2 and predicted max probability for zinc using Metal3D.

model is advantageous since metalloprotein design workflows require models to score the full continuum of zinc sites starting from a sub-optimal binding site only populated at high metal concentration to a highly organized zinc site in an enzyme with nanomolar metal affinity. The predicted probability can be used as a confidence metric or as an optimization target where mutations are made to increase probability of zinc binding.

## Site quality

The fraction of artifactual zinc binding sites in the PDB is estimated to be about $1/3$[52,53] similar to our test set used with 62% (119) well coordinated zinc sites with at least 2 distinct protein ligands and occupancy $> 0.5$. To reduce the amount of artifactual sites in the training set we presented the model with as many complete sites as possible by using crystal symmetry to add adjacent coordinating protein chains (e.g. 4HTM in Fig. S7). The frequency of artifacts in the training set is therefore much lower than 30%. The sites which still remain incomplete or that are wrongly modeled (e.g. 5ZZU in Fig. S7) and not excluded through the resolution cutoffs and filtering pro-cedures likely present only a small fraction of the training set and their signal is drowned out by the numerical superiority of the cor-rectly modeled sites. Deep learning models have been shown to be robust to noisy datasets[54] and Figure S7 highlights that Metal3D ignores issues with data quality (i.e. wrongly assigned metals or crystal artefacts it was trained on). If the model is used on artifactual sites or partially disordered ones it can still predict the metal location with high spatial accuracy but often indicates a lower confidence for the prediction (Figs. 2 and 5). For the identification of physiological sites a probability cutoff of $p > 0.75$ and the biological assembly of a structure should be used. For in crystallo sites the biological assembly and all symmetry adjacent structures and the same cutoff should be used.

Metal ion locators that rely on homology such as MIB perform worse on partial binding sites because reducing the quality of the available templates by including 1- or 2-coordinate sites would yield many false positives (similar to including less homologous structures for the template search). The deep learning based Metal3D can likely circumvent this because it does not require any engineered features to predict the location of the metal and learns directly from a full representation of the environment surrounding the binding site. This allows looking at low confidence sites in the context of a given environment.

## Influence of non-protein ligands

Exogenous ligands play an important role for metals in biology as all empty coordination sites of metals are filled with water molecules in case there is no other exogenous ligand with higher affinity present (e.g. a thiol). Like other predictors, both Metal1D and Metal3D do not consider water molecules or other ligands in the input as the quality of ligand molecules in the PDB varies[39,55]. In addition, other potential sources of input such as AlphaFold do not provide explicit waters wherefore models should not rely on water as an input source. It is also not possible to use in silico water predictions because common water placement algorithms to place deep waters[39,56,57] either rely on metal ions being present in the input or ignore them completely. Moreover, in protein design algorithms, water is usually only implicitly modeled (e.g. in Rosetta).

For Metal3D, the input channel that encodes the total heavy atom density also encodes an implicit water density where all empty space can be interpreted as the solvent. For Metal1D, the contribution of water molecules is considered in an implicit way when the score is assigned to a site by considering coordinations including water com-patible with the one observed (e.g. a $HIS_3Wat$ site is equivalent to a $His_3$ site for the scoring).

## Choice of architecture

This work is the first to report a modern deep learning based model destined for identification of metal ligands in proteins. Similar approaches have been used in the more general field of protein-ligand docking where a variety of architectures and representations have been used. 3D CNN based approaches such as LigVoxel[37] and DeepSite[36] commonly use a resolution of 1 Å and similar input features as our model to predict the ligand density. However, predicting the density of a multi-atomic ligand is more complex than predicting the density of mononuclear metal ions. We therefore did not deem it necessary to include a conditioning on how many zinc ions are present in the box and rather chose to reflect this in the training data where the model needs to learn that only about half of the environments it sees contain one or more zincs. This choice is validated by the fact that the output probability densities at sufficiently high probability cutoffs are spherical with their radius approximately matching the van der Waals radius of zinc. For multi-nuclear sites the densities are also well reproduced but the clustering step requires higher probability cutoffs to separate the densities for individual ions (Fig. S8).

Mesh convolutional neural networks trained on a protein surface representation[35] also have been used to predict the location and identity of protein ligands but this approach can only label the regions of the surface that bind the metal ion and is conceptually not able to return the exact location of the metal. Some metal ion binding sites are also heavily buried inside proteins as they mediate structural stability rendering them inaccessible to a surface based approach. The most recent approaches such as EquiBind[38] use equivariant neural networks such as En-Transformer[58] to predict binding keypoints (defined as 1/2 distance between the $C\alpha$ of the binding residue and a ligand atom). Explicit side chains are still too expensive for such models and these models assume a fixed known stoichiometry of the protein and ligand. Metal3D can also deal with proteins that do not bind a zinc and does not assume that the amount of ions is known. The lack of explicit side chain information renders equivariant models unsuitable for the design of complex metal ion binding sites supported by an intricate network of hydrogen bonds that need to be positioned with sub-angstrom accuracy. The framework of our model in contrast is less data- and compute-efficient than approaches representing the protein as graph due to the need to voxelize the input and provide different rotations of the input environment in training but the overall proces-sing time for our model is still low taking typically 25 seconds for a 250 residue protein on a multicore GPU workstation (20 CPUs, GTX2070). Sequence based models[59,60] can only use coevolution signals to infer

residues in spatial proximity that can bind a metal. This might be difficult when it comes to ranking similar amino acids such as aspartate and glutamate or even ranking different rotamers where sub-angstrom level precision is needed to identify the mutant with the highest affinity for zinc.

## Selectivity

In terms of selectivity, Metal3D has a clear preference for transition metals over main group metals after having been trained exclusively on zinc binding sites with recall and precision highest for zinc (Fig. 4B). For Metal3D environments that do not bind zinc but other metals are sampled as non-zinc binding and the method theoretically can learn to distinguish zinc from non-zinc metal sites. However, this is not the case and the method also predicts other metals (Fig. 4B) with high recall and precision. We attribute this to the general promiscuity of transition metal ion binding[61] in that most proteins select the metal they bind according to the Irving-Williams series in competitive binding conditions with metal selectivity in general not enforced by the binding site itself but rather by external factors such as compartmentalization or metallochaperones[61]. The only sites that Metal3D identifies for non-transition metals are the ones that have at least partial side chain coordination. Many sodium and potassium sites are using backbone carbonyl coordination exclusively, which is not common for zinc and those sites are therefore not detected even if they were included in the training data due to wrong labeling in the PDB (e.g. 5ZZU Fig. S7). The high recall for most other transition metals is therefore related to the fact that those binding sites have sidechains in similar conformation compared to zinc sites. Metal3D could be rapidly adapted to predict not only location but also the identity of the metal similar to recent work by Mohamadi et al.[62]. In the framework of Metal3D a semantic metal prediction would be possible where the same model predicts different output channels for each metal it was trained on. To achieve perfect selectivity using such a model will be difficult because sometimes non-native metals are used for crystallization experiments and most other transition metals have less structures available. For Metal1D selectivity will be harder to include in the method without modification as coordination environment (the only trainable parameter of Metal1D) is only somewhat selective toward zinc. In this work, we chose to work exclusively with zinc because it is the most redox stable transition metal and because many training examples are available and establish a conceptual framework how selectivity could be included showing that the implicit way of training on zinc and non-zinc environments is not enough to enforce strict zinc selectivity.

## Application for protein design

Protein design using 3DCNNs trained on residue identity has been successfully demonstrated and we anticipate that our model could be seamlessly integrated into such a workflow[31] to enable fully deep learning based design of metalloproteins. We are currently also investigating the combination of Metal3D combined with a classic energy-based genetic algorithm-based optimization to make design of metalloproteins[17] easier without having to explicitly model the metal to compute the stability of the protein. As the model computes a probability density per residue it can be readily integrated into established software like Rosetta relying on rotamer sampling.

The HCA2 application demonstrates the utility of Metal3D for protein engineering (Fig. 2). The thermodynamics of metal ion binding to proteins are complicated[63] and there are currently no high-throughput based experimental approaches that could generate a dataset large enough to train a model directly on predicting $K_d$. The data we use were obtained from a colorimetric assay with very high affinity of zinc in the picomolar range[64–68]. More recent studies using ITC[63] instead of the colorimetric assay indicate lower $K_d$ values in the nanomolar range for wild type HCA2. We can therefore only use the colorimetric data to estimate how well the model can recapitulate relative changes in the $K_d$ for different mutations in the first and second shell of a prototypical metalloprotein.

Metal3D allows moving away from using rational approaches such as the i, i + 4 di His motifs used for the assembly and stabilization of metalloproteins to a fully automated approach where potential metal binding configurations can be scored computationally[69–71].

## Metal3D vs. other methods

Metal1D is inferior to Metal3D for the prediction of metal ion binding sites because it produces more false positives while at the same time detecting fewer metal sites. Also, the positioning of sites is somewhat imprecise. This demonstrates the inherent limitation of using solely distance based features for prediction of metal location. BioMetAll which is the tool most similar to Metal1D also suffers from many false positive predictions with even worse performance compared to Metal1D. In contrast, Metal1D is more data-efficient than Metal3D and provides predictions faster. For large structural databases Metal1D could be run as a prefilter step to then provide high-accuracy predictions using Metal3D. While the MIB method produced decent results when a high template cutoff is used, for the updated MIB2 tool, we find no systematic improvement with only slightly higher recall even if the template database for zinc was extended from 499 to 2446 templates. MIB no longer is available and for MIB2 high-throughput analysis is not possible since a standalone or source code is not available and the webserver blocks multiple concurrent jobs. Metal3D is therefore the only tool that can provide high-quality interpretable predictions in reasonable time (ca. 25 seconds on a GPU workstation for a 250 aa protein). Metal1D while not as accurate as Metal3D is very fast and can be applied to large structural databases. While Metal3D currently is trained only on zinc it offers detection capabilities also for other transition metals at slightly lower recall and precision and the framework of the method could be readily extended to also provide identity of the predicted locations similar to recent work by Mohamadi et al.[62]. We therefore anticipate different applications for Metal3D such as protein-function annotation of structures predicted using AlphaFold2[72], integration in protein design software and detection of cryptic metal binding sites that can be used to engineer PPIs. Such cryptic metal ion binding sites in common drug targets could also be used to engineer novel metallodrugs. Many of these applications will allow us to explore the still vastly untapped potential of proteins as large multi-dentate metal ligands with programmable surfaces.

# Methods
## Dataset

The input PDB files for training were obtained from the RCSB[73] protein databank (downloaded 5th March 2021). We use a clustering of the structures at 30% sequence identity using mmseqs2[74] to largely remove sequence and structural redundancy in the input dataset. For each cluster, we check whether a zinc is contained in one of the structures, whether the resolution of these structures is better than 2.5 Å, if the experimental method is x-ray crystallography and whether the structure does not contain nucleic acids. If there are multiple structures fulfilling these criteria, the highest resolution structure is used. All structures larger than 3000 residues are discarded. We always use the first biological assembly to sample the training environments. The structures were stripped of all exogenous ligands except for zinc. If there are multiple models with e.g. alternative residue conformations for a given structure, the first one is used. For each biological assembly we used the symmetry of the asymmetric unit to generate a protein structure that contains all neighboring copies of the protein in the crystal such that metal sites at crystal contacts are fully coordinated. Statistics of the training and test set are provided in Figures S9-19.

The train/val/test split was performed based on sequence identity using `easy-search` in mmseqs2. All proteins that had no (partial) sequence overlap with any other protein in the dataset were put into the test/val set (85 proteins) which we further split into a test set of 59 structures and a validation set of 26 structures. The training set contained 2085 structures. (Supplemental Data S1).

For the analysis, we always used the biological assembly and not the symmetry augmented structure. For the selectivity analysis with respect to other metals, clusters from the PDB were randomly sampled to extract 100 biological assemblies per metal except for FE2 (57), NI (93), CU (68) and CO (30).

By default all zinc sites in the test and validation set were used for the analysis. Since some of the sites might be affected by the crystallization conditions, we also created a subset of all sites that contained at least 2 amino acid ligands to largely exclude crystallization artifacts. To analyze metal ion selectivity, we selected sites with at least 3 unique protein ligands to only use biologically significant sites with a high degree of metal preorganization as such sites should exhibit more selectivity for specific metals compared to sites with only 2 unique protein-ligands. For both categories we excluded metal ions that had occupancy < = 0.5

## Metal 1D

Metal1D uses a probability map derived from LINK records in protein structures (Fig. 1). The LINK section of a PDB file specifies the connectivity between zinc (or any other ligand) and the amino acids of the protein, and each LINK record specifies one linkage. This is an extension of the approach by Barber-Zucker et al.[75], in which LINK records were used to investigate the propensity of transition metals to bind different amino acids.

Using the training set we generated a probability map for the propensity of different coordination environments to bind a zinc (e.g. CCCC, CCHH etc.). For each zinc ion the coordination is extracted from the LINK records (Fig. 7A) excluding records involving only single amino acids (weak binding sites). Information in the LINK records of each PDB file are converted into a unique coordination environment by associating one letter code to each amino acid with a LINK with a zinc ion and alphabetically sorting this code. This ensures coordination environments such as CCH and CHC to be considered as equal. Also, LINK records containing water molecules are excluded because of the difficulties in placing water molecules a posteriori in 3D structures when metal ions are present and because data quality of modelled water molecules varies. The probability map contains the counts of coordination environments found and is generated from a list of pdb files, the training set in this case. A jupyter notebook is made available to be able to generate a probability map from a different set of pdb files (ProbMapGenerator.ipynb).

Making a prediction using Metal1D consists of two main steps (Fig. 7B): Identification of possible metal coordinating residues in the structure via a residue scoring step, and the scoring of putative sites, placed between the identified coordinating residues.

The protein structure is analyzed using the BioPandas python library[76]. To identify coordinating residues, a per residue score is assigned by performing a geometrical search from a reference point, defined as the coordinate of the most probable metal binding atom, within a search radius considered as roughly twice the typical distance between the metal ion and the binding atom of amino acids in proteins ($2.2 \pm 0.2$ Å as determined from LINK records). The search radius used was 5.5 Å in order to be able to take into account also deviations from the ideal coordination. In the case of amino acids which present more than one putative coordinating atom, such as e.g. histidine, the mid-point between the donor atoms is used as reference point and the search radius is enlarged accordingly. The atoms used as reference points for each amino acid and the increase in the search radius are reported in Supplemental Table S1. The score is assigned to each

amino acid considering all the other reference points of other amino acids within the search radius, and summing the probabilities in the probability map for coordinations compatible with the one observed. In the ideal case, a score of 1 corresponds to an amino acid surrounded by all possible coordinating amino acids observed in the probability map. In practice, scores result between 0 and <1. Once all amino acids in the chain are scored, the metal location predictions are made grouping the highest-scored amino acids in clusters (defined as the ones within the chosen threshold, i.e. the t parameter, with respect to the highest-scored one) based on distance. This is done using `scipy.spatial.distance_matrix` and grouping together highest-scored amino acids closer than twice the search radius. For each cluster, a putative site is located in space as a weighted average between the coordinates of the reference point of each amino acid, using as weighting factor the amino acid score. For a given cluster of $N$ high scored residues (with xyz locations $\{\mathbf{r}_1, ..., \mathbf{r}_N\}$ and scores $\{score_1, ..., score_N\}$), the xyz location of the predicted site ($\mathbf{r}_{site}$) is computed as

$$\mathbf{r}_{site} = \frac{\sum_{i=1}^{N} score_i \times \mathbf{r}_i}{\sum_{i=1}^{N} score_i} \qquad (1)$$

For isolated amino acids with a high score (e.g. a single histidine) the same score is assigned to the closest reference point from another amino acid, to be able to compute the position of the metal as before, i.e. using a weighted average. In this case the metal will be placed at the midpoint between the highest scored residue (the single element of the cluster) and the amino acid to which the fictitious score is assigned. Possible artifacts resulting from this fictitious score are resolved in the final step of the prediction.

After the putative site has been placed, a score is assigned by performing a geometrical search centered on the predicted metal coordinates (within 60% of the search radius, i.e 3.3 Å) and a final score is now assigned to the site. The final score is assigned in the same way as the amino acid scores based on the probability map, and has the advantage of being able to sort the predicted metal sites based on their frequency in the training set. A cutoff parameter, by default equal to the cutoff used for amino acid scoring (i.e. the t parameter), is used to exclude sites with a probability lower than a certain threshold with respect to the highest-scored one. This final scoring also mitigates the errors which can be introduced by calculating the coordinates of the site simply as a weighted average excluding or assigning a low probability to the site ending in unfavorable positions in space.

## Metal 3D

**Voxelization.** We used the moleculekit python library[37,77] to voxelize the input structures into 3D grids. 8 different input channels are used: aromatic, hydrophobic, positive ionizable, negative ionizable, hbond donor, hbond acceptor, occupancy, and metal ion binding site chain (Fig. 8, Supplemental Table S2). The channels are assigned using AutoDockVina atom names and a boolean mask. For each atom matching one of the categories a pair correlation function centered on the atom is used to assign the voxel value[37]. For the target tensor only the zinc ions were used for the voxelization. The target tensor was discretized setting any voxel above 0.05 to 1 (true location of zinc), all other to 0 (no zinc). We used a box size of 16 Å centered on the C$\alpha$ atom of a residue, rotating each environment randomly for training before voxelization. The voxel grid used a 0.5 Å resolution for the input and target tensors. Any alternative side chain conformations modeled were discarded keeping only the highest occupancy. For the voxelization only heavy atoms were used. For all structures selected for the respective sets we partitioned the residues of the protein into residues within 12 Å of a zinc ion and those further away (based on the distance to the C$\alpha$ atom). A single zinc site will therefore be present

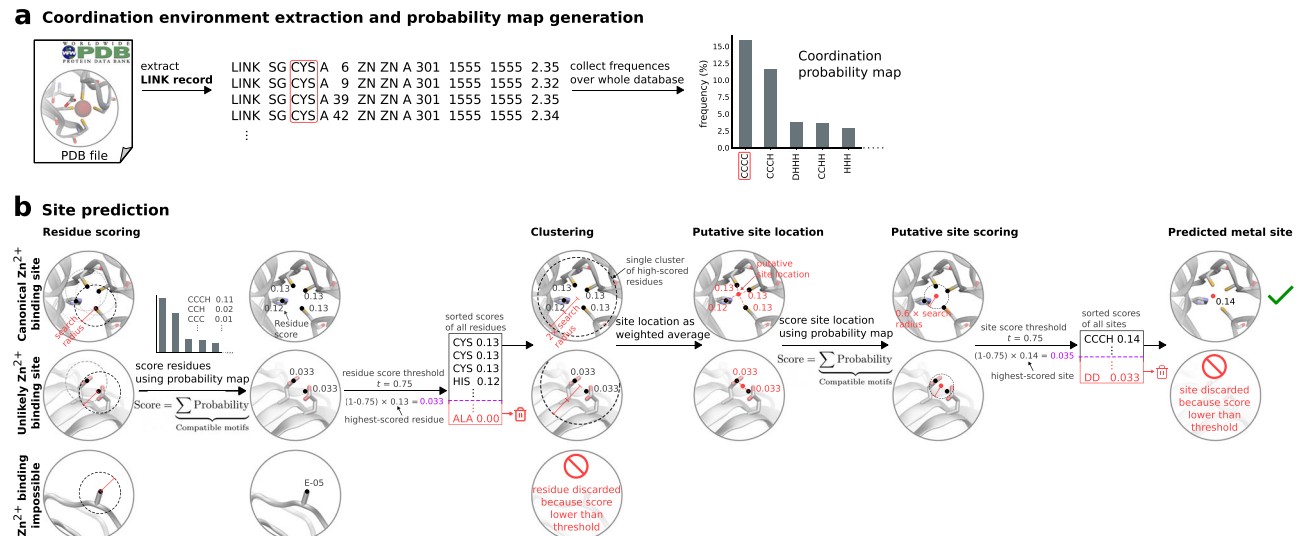

**Fig. 7 | Metal1D method. a** Coordination environment extraction from `LINK` records of PDB files and probability map generation **b** Site prediction, showing examples for a Zn²⁺ binding site, correctly predicted, and unlikely Zn²⁺ binding site, discarded in the putative site scoring step because of low probability of the metal site, and an amino acid for which Zn²⁺ binding is impossible, discarded in the residue scoring step because of low score of the amino acid.

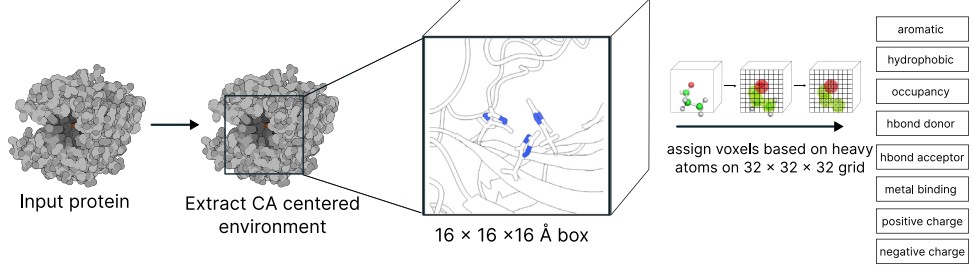

**Fig. 8 | Voxelization of an environment.** A Cα centered box is voxelized using moleculekit library. For each atom an atom centered pair correlation function is used to assign voxel values in 8 different channels (see Table S5) for atom types.

many times in the dataset but each time translated and rotated in the box. A balanced set of examples was used sampling equal numbers of residues that are close to a zinc and residues randomly drawn from the non-zinc binding residues. The sampling of residues is based on the biological assembly of the protein, the voxelization is based on the full 3D structure including neighboring asymmetric units in the crystal structure. The environments are precomputed and stored using lxf compression in HDF5 files for concurrent access during training. In total, 252324 environments were voxelized for the training set, 6550 for the test set, 3067 for the validation set. The voxelization was implemented using ray[78].

**Model training.** We used PyTorch 1.10[79] to train the model (Fig. 9). All layers of the network are convolutional layers with filter size 1.5 Å except for the fifth layer (Conv5 in Fig. 9) where a 8 Å filter is used to capture long range interactions. We use zero padding to keep the size of the boxes constant. Models were trained on a workstation with NVIDIA GTX3090 GPU and 32 CPU cores. Binary Cross Entropy[80] loss is used to train the model. The rectified linear unit (ReLU) non-linearity is used except for the last layer which uses a sigmoid function that yields the probability for zinc per voxel. A dropout layer ($p = 0.1$) was used between the 5th and 6th layers. The network was trained using Ada-Delta employing a stepped learning rate (lr = 0.5, γ = 0.9), a batch size of 150, and 12 epochs to train.

**Hyperparameter tuning.** We used the ray[tune] library[78] to perform a hyperparameter search choosing 20 different combinations between the following parameters with the best combination of parameters in bold.

- filtersize: **3**,4 (in units of 0.5 Å)
- dropout : **0.1**, 0.2, 0.4, 0.5
- learning rate : **0.5**, 1.0, 2.0
- gamma: 0.5, 0.7, 0.8, **0.9**
- largest dimension 80, 100, **120**

**Grid Averaging.** The model takes as input a `(8,32,32,32)` tensor and outputs a `(1,32,32,32)` tensor containing the probability density for zinc centered on the Cα atom of the input residue (last step in Fig. 9). Predictions for a complete protein were obtained by voxelizing select residues of the protein (default all cysteines, histidines, aspartates, glutamates), predicting them individually using the above described model and averaging the boxes using a global grid (Fig. 10). 98% of the metal sites in the training data have at least one of those residues closeby wherefore this significant decrease in computational cost seems appropriate for most uses. The global grid is obtained by computing the bounding box of all points and using a regular spaced (0.5 Å) grid. For each grid point in the global grid the predicted probability maps within 0.25 Å of the grid point are averaged. The search is sped up using the KD-Tree implementation in scipy[81].

**Metal ion placement.** The global probability density is used to per-form clustering of voxels above a certain probability threshold (default $p = 0.15$, cutoff 7 Å) using AgglomerativeClustering implemented in scikit-learn[82] (Fig. 10). For each cluster the weighted average of the voxels in the cluster is computed using the probabilities for each point

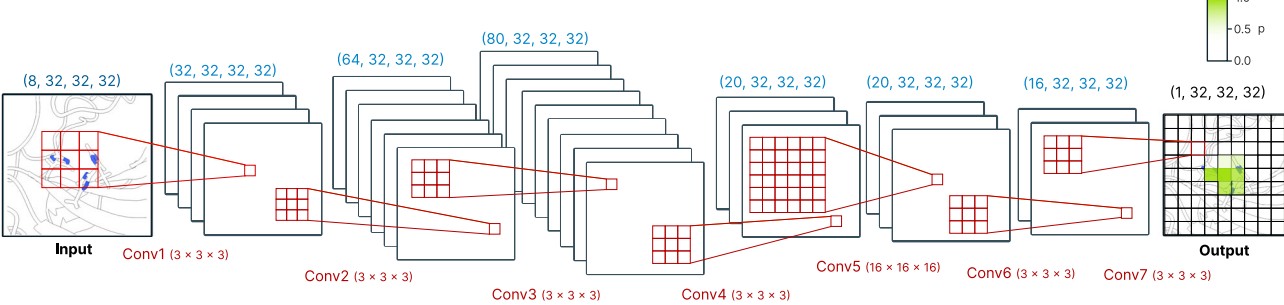

**Fig. 9 | Sketch of model.** Residue centered zinc densities are predicted based on input environment. Number of layers not to scale. Blue tensor sizes indicate number of channels and size of grid (always 32 × 32 × 32) Convolutional filters (red) are trained and extract information (3 × 3 × 3 convolutions) or aggregate information (16 × 16 × 16 convolution).

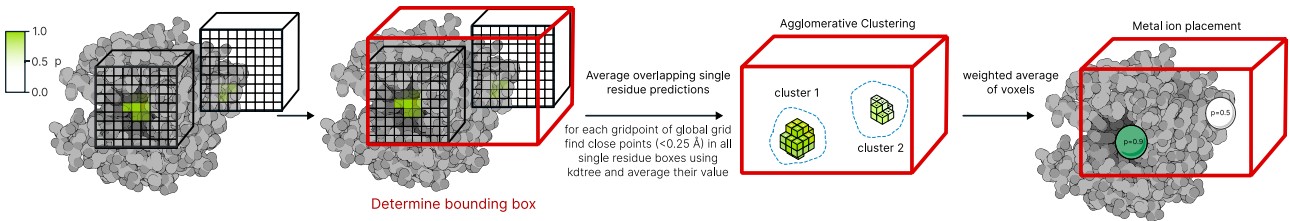

**Fig. 10 | Grid averaging and metal ion placement.** A bounding box for the global grid is defined based on all predictions and residue probability maps are aggregated using KD-Tree search. Ions are placed after AgglomerativeClustering and taking the weighted average of all voxels in a cluster. Probability is the maximum probability of the cluster.

as the weight. This results in one metal placed per cluster. For each placed ion the maximum probability of a voxel in a cluster is taken as the probability of the ion.

**Visualization.** We make available a command line program and interactive notebook allowing the user to visualize the results. The averaged probability map is stored as a `.cube` file. The most likely metal coordinates for use in subsequent processing are stored in a `.pdb` file. The command line program uses VMD[83] to visualize the input protein and the predicted density, for the jupyter notebook 3Dmol.js/py3Dmol[84] is used.

### Evaluation and Comparison
**MIB & MIB2.** We compared against the template based predictors MIB[21] (no longer available) and MIB2[22] (same as MIB but with extended template database) using the webserver located at http://combio.life. nctu.edu.tw/MIB2/. Structures from the testset were manually uploaded, the job ids saved and predictions extracted from the html output of the job. The t-scores for MIB were chosen based on the description of the method, for MIB2 no such recommended values were provided in the description of the method and we compared the distribution of t-scores choosing a set of t-scores approximately matching the old distribution of MIB (Fig. S6).

**BioMetAll.** Predictions were run using the standalone BioMetAll v1.0 programm obtained from https://biometall.readthedocs.io/en/latest/installation.html.

### Evaluation metric
In order to standardize the evaluation between different tools, we always used the same test set used for the training of Metal1D and Metal3D. In order to compute standard metrics such as precision and recall, we chose to assess the performance of all assessed tools (Metal1D, Metal3D, BioMetAll, MIB) in a binary fashion. Any prediction within 5 Å of an experimental metal site is counted as true positive (TP). Multiple predictions by the same tool for the same site are counted as 1

TP. Any experimental site that has no predicted metal within 5 Å is counted as false negative (FN). A false positive (FP) prediction is a prediction that is not within 5 Å of a zinc site and also not within 5 Å of any other false positive prediction. If two or more false positive predictions are within 5 Å, they are counted as a single false positive prediction for the same site. In practice, we first evaluate the true positive and false negative predictions and remove those from the set of predicted positions. The remaining predictions are all false positives and are clustered using AgglomerativeClustering with a radius of 5 Å. The number of false positives is determined from the number of clusters. Using the binary metric we assessed how good the models are at discovering sites and how much these predictions can be trusted.

In order to assess the quality of the predictions, we additionally compute for all the true positive predictions the mean of the Euclidean distance between the true and predicted site (mean absolute deviation MAD). For Metal1D, MIB, and BioMetAll, MAD was computed for all predictions above the threshold within 5 Å of a true zinc site where $\sum$ predictedsites $\geq \sum$ TP. This was done as some tools predict the same site for different residue combinations and we wanted to assess the general performance for all predicted sites above a certain cutoff and not just for the best predicted site above the cutoff. For Metal3D the weighted average of all voxels above the cutoff was used.

Precision was calculated as

$$\text{Precision} = \frac{\#\text{correct metal sites}}{\#\text{correct metal sites} + \#\text{false positive clustered}} = \frac{\text{TP}}{\text{TP} + \text{FP}} \quad (2)$$

Recall was calculated as

$$\text{Recall} = \frac{\#\text{correct metal sites}}{\#\text{correct metal sites} + \#\text{not found metal sites}} = \frac{\text{TP}}{\text{TP} + \text{FN}} \quad (3)$$

**Model assessment Metal3D.** To evaluate the trained models we monitored loss and how accurately the model predicts the metal

density of the test set. We used a discretized version of the Jaccard index setting each voxel either as 0 (no metal) or 1 (zinc present). We tested multiple different decision boundaries (0.5, 0.6, 0.75, 0.9) and also compared a slightly smaller centered box to remove any spurious density at the box edges, where the model has only incomplete information to make predictions.

The Jaccard index is computed as

$$J = \frac{\# V_p \cap V_{exp}}{\# V_p \cup V_{exp}} \qquad (4)$$

where $V_p$ is the array of voxels with predicted probability above the decision boundary and $V_{exp}$ is the array of voxels with the true metal locations also discretized at the same probability threshold.

**HCA2 mutants**. The data for human carbonic anhydrase 2 (HCA2) mutants was extracted from refs. [65–69] and the crystal structure 2CBA[48,49] was used. The zinc was modeled using the zinc cationic dummy model forcefield[51] and we verified that energy minimization produced the correct coordination environment. The Richardson rotamer library[85] was used with the EVOLVE-ddG energy function to compute the most stable rotamer for a given mutation with the zinc present. The lowest-energy mutant was used for the prediction of the location of metals using Metal3D.

**Reporting summary**
Further information on research design is available in the Nature Portfolio Reporting Summary linked to this article.

## Data availability
The list of PDB identifiers used to train and evaluate the models and data required to reproduce figures and tables in this manuscript have been deposited on Zenodo under https://doi.org/10.5281/zenodo.7015849.

## Code availability
Code is available under https://github.com/lcbc-epfl/metal-site-prediction[86] and also on Zenodo under https://doi.org/10.5281/zenodo.7015849[87]. EVOLVE v0.2 code is available on https://doi.org/10.5281/zenodo.5713801[88].

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

## Acknowledgements

Supported by Swiss National Science Foundation Grant Number 200020-185092 with computational resources from the Swiss National Computing Centre CSCS to U.R.

## Author contributions

S.L.D and A.L designed research, S.L.D, A.L, U.R conceptualized research, S.L.D and A.L developed methodology and software, S.L.D and A.L wrote first draft, S.L.D, A.L, U.R revised and edited draft, U.R supervised research and acquired funding.

## Competing interests

The authors declare no competing interests.
