## [Peer Review File · Nature Communications]

REVIEWER COMMENTS

Reviewer #1 (Remarks to the Author):

The work entitled “Accurate prediction of transition metal ion location via deep learning” proposed two machine learning methods which can be used to identify and locate zinc ions in given protein structures. Metal ion is one of the most important cofactors of protein, thus the prediction of metal locations is of great important in the study of the structure and function of proteins. The theories and methods used in this work is solid and the conclusion can be well support by the results. However, I still have several questions that need the authors to address:

1. As stated in the previous study (Nat. Protoc. 2014, 9, 156.), the structure of a large number of metalloproteins in problematic in the Protein Data Bank. Is it reasonable to take the resolution of X-ray experiment as the only criteria in the preparation of training set?
2. According to Figure S1 and Figure 4, the influence of $p(\text{metal})$ value on the performance of Metal3D is unobvious. However, as shown in Figure 3, the number of false positive probes is highly depended on the $p(\text{metal})$ value. Further explanations should be provided.
3. Although these two models are trained with zinc proteins, it seems that they can also detect the Ca^{2+} positions (Figure 4). Is it reasonable? As zinc and calcium prefer different kind of ligands.
4. In line 316, The authors states that “Both of our methods could be rapidly adapted to predict not only location but also the identity of the metal”. I think its overclaimed because these two method cannot distinguish ZN with MN and CO according to Figure 4 and Figure S3.
5. Can these methods deal with multi-core metal centers? Such as the NDM-1 protein which has two zinc ions in its active site.
6. More details should be provided in the method section and the current descriptions should be more clear. For example, its better to use a figure to show how to calculate the probability map, whish is very important for potential readers who want to reprocude these two methods.

Reviewer #2 (Remarks to the Author):

The authors present a method and a tool to predict transition metal sites in structures, via deep learning. The topic of the paper is challenging and their tool appears to perform better than the

available methods representing a step forward towards the achievement of a scientific goal that would allow a considerable advancement in the field of metalloproteins. However, I have some major concerns that I'm going to detail below.

- The tool presented in the paper is trained and tested on a zinc dataset. However, the authors claim that it can predict transition metal ions in general (as reported in the title of the paper). As far as I understand, the tool predicts the presence of a zinc ion also when a different metal ion is actually present in the protein. In a real case, the prediction would then mislead the user to consider as zinc-binding a protein that binds another metal ion. This lack of selectivity should be seen as a weakness rather than a strength of the tool. I think it would be more appropriate to state that the predictor is for zinc only.
- From the paper, it is not clear what Metal1D could be useful for; should it be used in a complementary way to Metal3D? If so, what advantage would it bring to the Metal3D user? Otherwise, I don't see the reason to present two independent tools, one of which is clearly better than the other.
- In my experience, the large majority of zinc sites composed by less than three amino acids are largely spurious (at least 60% of cases for sites with two endogenous ligands). The authors should clearly address this point, clarifying whether their tool is intended to identify physiological zinc sites, or any zinc site. In the former case, predicting spurious sites would decrease the precision of the program.
- In the Introduction, the authors state that many tools are available for prediction of metal sites in structures (they list seven different tools, lines 58-60), however they compare their performances with only two tools (MIB and BioMetAll). They should clarify the reason for this selection.
- The authors state that they improved the performance of predictions with respect to existing tools, but do not explicitly report the precision and the recall of their tool, for different p and t values. This would help the user in understanding the results. In this respect, the curve reporting the precision versus recall should be included in the main text of the article, not in the Supplementary Material.
- The Materials and Methods are quite difficult to read, especially for a non-specialized reader. A possible help could be to add a brief description of what the various pre-existing modules used in the program do.

Reviewer #3 (Remarks to the Author):

This manuscript presents the development of a new deep learning-based tool for the prediction of metal binding sites. It appears to have been thoroughly developed and addresses a difficult problem in an original way. I have the following comments:

1. As mentioned in the manuscript, metals are often present in the buffers used for crystallisation. What steps have been taken to ensure that the metal binding sites used in training and testing are biologically relevant?
2. I think the results would benefit from a brief explanation of the dataset i.e. how many binding sites are present in the test set and from how many different proteins?
3. On the dataset - reading the methods I get the impression that testing/validation was performed on a very small number of protein structures - clearly a large number of structures are required for training but to have only 85 proteins for testing, which are further split seems very small. Is the set that performance with other methods is compared? If so, then this small dataset could be biased and not represent performance of any of the methods on a larger dataset.
4. Greater clarity is needed on the presentation of the performance of the Metal1D and Metal3D, especially compared to other methods. Figure 3 is difficult to interpret and also makes extensive use of the combination of red and green. The text and this figure also refer to t , c and p as values but it is not clear what these are, this needs to be explained. The only one that appears to make sense is p , which appears to relate to probability. Given that these deep learning methods are likely to associate each prediction with a probability or confidence score, I would think it would be much easier to understand to present a precision-recall graph (present in supplementary figure 2) of the results rather than the difficult to interpret bar chart. Then looking at the precision-recall graph the performance is not that good - at a high probability threshold a recall of 40% is obtained with precision close to 1 (this is fairly good) but at the lowest threshold - recall only reaches 60% at precision of 40%.
5. Binding sites predicted with 5 angstroms of an actual binding site are classed as true positives - why was this distance selected, is it too lenient? What is the effect of reducing this threshold?
6. What is the purpose of the section on selectivity for other metals? This needs to be clearer. Why were only 25 structure selected - this is a small number of structures to test on, there must be many more available? . Figure 4B shows the recall for these biding sites but this is not very informative without precision as well. Again the manuscript should be presenting the performance in a way that is easy to interpret.

Reviewer #1 (Remarks to the Author):

The work entitled "Accurate prediction of transition metal ion location via deep learning" proposed two machine learning methods which can be used to identify and locate zinc ions in given protein structures. Metal ion is one of the most important cofactors of protein, thus the prediction of metal locations is of great important in the study of the structure and function of proteins. The theories and methods used in this work is solid and the conclusion can be well support by the results. However, I still have several questions that need the authors to address:

1. As stated in the previous study (Nat. Protoc. 2014, 9, 156.), the structure of a large number of metalloproteins in problematic in the Protein Data Bank. Is it reasonable to take the resolution of X-ray experiment as the only criteria in the preparation of training set?

We thank the reviewer for raising this important point. We were also concerned about data quality when preparing the data however there are two important points that we think contribute to the success of the method despite some noisy data in the training set.

First, we believe that setting resolution cutoff to 2.5 Å, always using the highest resolution structure per cluster that contains zinc (effective resolution <2.5 Å, see statistics added in Supplemental Figure S17) and only using X-Ray structures ensures that data quality is somewhat consistent for the good sites with random perturbations for wrong metal ions (e.g wrong identity, bad geometry etc.). Figure 1 in Nat. Protoc. 2014, 9, 156. does not indicate how resolution dependent bad metal sites are. doi: [10.1021/ic401072d](https://doi.org/10.1021/ic401072d) shows that e.g bond deviations drastically increase above 2.5 Å highlighting that the resolution cutoff reduces the amount of bad sites by a good margin. In addition, many sites that are artefacts (in the test set we have about 30% artefacts) are restored to full coordination via symmetry augmentation in our training data. The model therefore sees mostly well organized *in crystallo* metal binding sites.

Secondly, deep learning based methods (as opposed to classical machine learning) have been shown to be very robust to noisy datasets. As an example, an image classification model with 2D CNNs trained on a dataset where for each correctly labelled sample 100 randomly labelled samples are added to the dataset can still achieve 90% accuracy on the test set (<https://arxiv.org/pdf/1705.10694.pdf>). Here, the ratio of correct training examples to wrong examples is much more favorable. Metal3D therefore simply learns to ignore the data quality issues in the dataset because the signal from the correct metal sites is much stronger. We have picked a few examples from the training set and the referenced paper that are wrongly modelled and show them in Supplemental Figures S7 and S8 highlighting that the model takes context (biological assembly vs. crystal environment) into account and ignores bad training examples (e.g wrong metal modelled like in 5ZZU). We have also expanded the discussion on site quality to better reflect this.

Supplemental Figure S7: Performance of Metal3D on wrong metal sites: **1QY6:** Deposited structure contains a Zn²⁺ site at a crystal contact wrongly assigned as K⁺. Metal3D only identifies the site as zinc site with high probability if the symmetry mates are present. **4EVB :** Metal3D was explicitly trained on all the Zn²⁺ sites in this structure. In inference, the site with occupancy 0.25 is not predicted showing that Metal3D ignores issues with data quality in the training set. **5ZZU :** An alkali/earth-alkali binding site (e.g NA, CA or MG according to CheckMyMetal, CheckMyBlob) wrongly assigned as Zn²⁺. In inference, Metal3D does not predict this site even though it was trained on this site. **4HTM** All metal ion binding sites in this biological assembly are crystal contacts. Metal3D was trained on this structure and only predicts the metal ion binding site that has 2 protein ligands using the biological assembly. When symmetry mates are present the other sites with coordination partners from symmetry adjacent structures are also predicted. Biological assemblies in blue, symmetry mates in green, experimental zinc sites as purple, predicted locations blue. Occupancy (occ=) and maximum predicted probability (p=) indicated for select sites.

2. According to Figure S1 and Figure 4, the influence of p(metal) value on the performance of Metal3D is unobvious. However, as shown in Figure 3, the number of false positive probes is highly depended on the p(metal) value. Further explanations should be provided.

The mean absolute deviation in Figure 4 can only be computed if we have an experimental location of the zinc ion and a predicted location available (referred to as true positive(TP) throughout the text). For non-identified binding sites (false negative FN) or false positive sites there is no possibility to compute a deviation as only one location is available. We have made clearer in the caption of Figure 4A and in the text that the MAD metric is only computed using the true positive sites. We now provide a precision-recall curve in Figure 3 to help the reader judge recall and precision in relation to p(metal) and Figure 4A to judge the quality of the positioned metal ions as the counting in Figure 3 is binary only (is the predicted zinc ion within 5Å of the experimental location yes or no). It makes the model more useful that the p(metal) cutoff changes the FP and TP rate but does not affect the spatial precision of the location predictions.

Figure 4: Mean absolute deviation of correctly predicted sites and selectivity for other ions

A For all sites where predicted and experimental locations are available (true positives in Figure 3) we compute the mean absolute deviation (MAD) of predicted zinc ion locations using Metal1D, Metal3D, BioMetAll and MIB on the test set. Some tools predict multiple locations (n) per site (TP). *For MIB and MIB2 we used 2 structures less because the server did not accept these structures. For each tool the whisker plot indicates the median (white dot) and the first quartiles (black box). **B** Precision and recall for the test set and 100 randomly drawn structures (93 NI, 68 CU, 57 FE2, CO 30) for other transition (blue), earth-alkali (pink) and alkali (brown) metals. Metal1D as light squares, Metal3D as circles. Threshold used for Metal3D p= 0.75, threshold used for Metal1D t=0.75.

3. Although these two models are trained with zinc proteins, it seems that they can also detect the Ca²⁺ positions (Figure 4). Is it reasonable? As zinc and calcium prefer different kind of ligands.

As we show in Figure 4B, precision and recall for alkali and earth-alkali ions is lower than for transition metals with higher distance deviations (Figure S2) and lower probability of the sites (Figure S3). As suggested by reviewer 3, we have expanded the set of structures for the selectivity analysis and show that for Metal3D, recall is only high for transition metals and lower (<0.4) for alkali and earth-alkali metals. Currently, the model has only implicitly been trained to be selective for metals in that we have shown the model a balanced number of environments that contain zinc and that do not contain zinc. The latter environments can contain other metals but they are not explicitly present in the input so the model does not see the metal but it sees the preorganized sidechains for those sites which theoretically should impart it with some selectivity towards zinc. Given that calcium and magnesium sites often use sidechain coordination by aspartate and glutamate with similar coordination distances (e.g Asp - Zn²⁺ 2.17 ± 0.2 Å, Asp - Ca²⁺ 2.5 ± 1.0 Å [doi:10.1021/ic401072d](https://doi.org/10.1021/ic401072d) , [doi:10.1002/prot.10093](https://doi.org/10.1002/prot.10093))) it is reasonable to assume that the model would recognize them as zinc binding if they adopt similar conformation to aspartate and glutamate sidechains in zinc binding sites. It is not uncommon for zinc sites to contain one or multiple aspartates or glutamate residues (e.g DHHH, EHH, DHH, DDHH in Figure S19) . Alkali metals which use backbone coordination much more frequently and have different size compared to transition metals are mostly not predicted by the model (recall ~5%) and thus match the inductive prior.

4. In line 316, The authors states that “Both of our methods could be rapidly adapted to predict not only location but also the identity of the metal”. I think its overclaimed because these two method cannot distinguish ZN with MN and CO according to Figure 4 and Figure S3.

This is an important point and relates to the same issue raised in comment 3. At the moment during training of Metal3D selectivity is not enforced explicitly and due to the promiscuous nature of metal binding according to the Irving Williams series it is not surprising that the model identifies sites that bind Mn or Co because most of those sites would actually be filled with zinc in a solution of the protein and equal concentrations of zinc and the native metal because the zinc complex is more stable than the native metal (except for native Ni²⁺ & Cu²⁺ binding sites). The binding sites of other first-row transition metals are different in composition and architecture but not so significantly that they will not bind zinc for most sites since ionic radii and other properties are somewhat similar. For alkali ions on the other hand, the binding modes are very different and clearly the model does misidentify only very few of them.

In order to fix this problem of misidentifying other transition metal ions one could either modify the training data to include more other metal ion binding sites as negative examples (currently they are only randomly sampled together with all non-metal environments) or force the network to not only predict zinc location but different densities for each input metal.

A recent paper from September 2022 (ref) confirms that this is possible using the exact same input we use with a different output (classification 3DCNN instead of fully convolutional 3DCNN) That network can predict the identity of the metal. In this work, we intend to present a general strategy to location prediction for metal ions (in this case applied to zinc) using deep learning. We have updated the title to reflect this: *Metal3D: A general deep learning framework for accurate metal ion location prediction in proteins*

We agree that for the other method proposed here (Metal1D) it is less clear how selectivity can be enforced without modifying the method with a meta-predictor given that the only input is the coordination environment. With the new set of up to 100 structures per metal Metal1D is more selective for zinc than with the smaller set indicating that the coordination environment as sole input is somewhat selective towards specific metals.

5. Can these methods deal with multi-core metal centers? Such as the NDM-1 protein which has two zinc ions in its active site.

Most zinc sites in our manuscript are mono-nuclear metal centers. We have now added information regarding multi-core metal centers in the results (reproduced below), discussion and supplemental information (Figures S8, S9, S10).

We assessed Metal3D for its performance on multi-nuclear metal sites and also collected statistics on their prevalence in training and test sets (Figure S8 and S9). For the di-nuclear NDM-1 protein (PDB 4EYL), Metal3D produces a density map that well reproduces both metal ions (His₃, HisAspCys) which are separated by 4.0Å. There is a third spurious prediction in the vicinity of the active site with no experimental support for metal binding in the structure. This site has a realistic coordination motif (HisGluAsp) and $p=0.74$, which is higher than the probability predicted for the HisAspCys site $p=0.66$. The clustering which places individual probes in the density map works well for the metal that is coordinated by His₃ but not for the other which is coordinated by HisAspCys. At higher isovalues for the probability density, increased support is present for the His₃ motif (max $p=0.97$) compared to the other motif (max $p = 0.66$). With the default settings (clustering threshold 0.15, distance threshold 7 Å) only one probe is placed. Two probes can be obtained by setting the clustering threshold to 0.5 separating the probability densities for each metal site.

Two other examples were extracted from the training set after literature review [doi:10.1093/jn/130.5.1437S](https://doi.org/10.1093/jn/130.5.1437S): alkalinephosphatase and phospholipase C.

Both enzymes have tri-nuclear metal centers and were contained in the training set. For phospholipase C (1AH7), Metal3D correctly identifies 2 out of 3 metal sites. The metal site

with one backbone coordinating amino acid has some density extending to the identified metal site close to it (3.5 Å) but not enough to place a separate zinc there after clustering. For alkaline phosphatase we analyzed a structure (1ALK) that was not directly contained in the training set which contained two Zn²⁺ and one Mg²⁺ ion. The structure contained in the training set (5C66) instead has three Zn²⁺ modelled. Metal3D correctly identifies the two Zn²⁺ modelled in 1ALK but not the magnesium even though it was trained on it in 5C66 containing a Zn²⁺ in this position. The site has a threonine coordinating the ion, which is not common for zinc.

Supplemental Figure S8: Multi-nuclear metal sites. **4EYL:** Metal3D probability density has density for the two modelled zinc ions and one FP. The probability density separates into two separate density blobs with $p > 0.5$. For placing ions using clustering higher probability cutoff $p > 0.5$ is required. **1AH7:** Two of 3 experimental zincs have probability density predicted by Metal3D. One ion with backbone coordination is not found. **1ALK:** The two zinc ion binding sites are correctly found by Metal3D with correctly placed probes at default cutoff. The Mg²⁺ site which can also bind Zn²⁺ (as is the case for 5c66 in the training set) is not predicted even though the model was trained on it. Ions placed by Metal3D after clustering voxels in dark blue. Experimental Zinc in blue olive.

6. More details should be provided in the method section and the current descriptions should be more clear. For example, its better to use a figure to show how to calculate the probability map, which is very important for potential readers who want to reproduce these two methods.

Several of the reviewers have raised this point and we have made improvements to the legibility of the methods section and added multiple figures to better describe Metal1D (probability map generation and scoring steps) and Metal3D (voxelization, architecture, grid averaging) in Figures 7 - 10 . Code for the algorithms we describe is available on github and we added a jupyter notebook where the user, starting from a list of PDB entries, can extract the coordination motifs from LINK records and generate a probability map. (ProbMapGenerator.ipynb). This is saved as a csv file which can be uploaded to the Colab notebook already available and used instead of the one provided (probability map for Zn²⁺).

A Coordination environment extraction and probability map generation

B Site prediction

Figure 7. Metal1D method A Coordination environment extraction from LINK records of PDB files and probability map generation **B** Site prediction, showing examples for a Zn^{2+} binding site, correctly predicted, an unlikely Zn^{2+} binding site, discarded in the putative site scoring step because of low probability of the metal site, and an amino acid for which Zn^{2+} binding is impossible, discarded in the residue scoring step because of low score of the amino acid.

Figure 8. Voxelization of an environment: A $C\alpha$ centered box is voxelized using moleculekit library. For each atom an atom centered pair correlation function is used to assign voxel values in 8 different channels (see Table S2)

Figure 9. Sketch of model that predicts residue centered metal densities based on input environment. Number of layers not to scale. Blue tensor sizes indicate number of channels and size of grid (always $32 \times 32 \times 32$) Convolutional filters (red) are trained and extract information ($3 \times 3 \times 3$ convolutions) or aggregate information ($20 \times 20 \times 20$ convolution)

Figure 10. Grid averaging and metal ion placement: A bounding box for the global grid is defined based on all predictions and residue probability maps are aggregated using KD-Tree search. Ions are placed after Agglomerative Clustering and taking the weighted average of all voxels in a cluster. Probability is the maximum probability of the cluster.

Reviewer #2 (Remarks to the Author):

The authors present a method and a tool to predict transition metal sites in structures, via deep learning. The topic of the paper is challenging and their tool appears to perform better than the available methods representing a step forward towards the achievement of a scientific goal that would allow a considerable advancement in the field of metalloproteins. However, I have some major concerns that I'm going to detail below.

1. The tool presented in the paper is trained and tested on a zinc dataset. However, the authors claim that it can predict transition metal ions in general (as reported in the title of the paper). As far as I understand, the tool predicts the presence of a zinc ion also when a different metal ion is actually present in the protein. In a real case, the prediction would then mislead the user to consider as zinc-binding a protein that binds another metal ion. This lack of selectivity should be seen as a weakness rather than a strength of the tool. I think it would be more appropriate to state that the predictor is for zinc only.

This is an important point and in line with the concerns of reviewer 1 comments 3 & 4. We have made changes to the text and title to address this concern.

In brief: Metal3D was only *implicitly* trained to be selective towards zinc and the framework of the tool is readily extensible to other metals. We tested/assessed selectivity with respect to other transition metals as well as alkali and earth alkali metals, showing that Metal3D in its current version is not a selective zinc predictor. However, without having been explicitly trained for selectivity, the model is already very selective against alkali metals and somewhat selective towards earth alkali metals identifying only those sites with similar coordination motifs to zinc. With respect to other transition metal ions, it can be expected (based on how transition metal selectivity is mainly enforced in nature, i.e. through compartmentalization or metal chaperones (ref1, ref2, ref3)) that most Zn, Co, Mn, Fe, Cu binding sites can bind zinc if only zinc is available and bind zinc according to the Irving Williams series if the native metal and zinc is available (see e.g. ref4, ref5). There is of course some difference in the coordination motifs of these metals (number of ligands, amino acid frequency) but in general zinc can bind in binding sites for tetrahedral or octahedral geometries since it has a full d10 shell and therefore no implicit preference for a certain coordination number.

Recent work by Mohamadi et. al. (ref6) shows that 3DCNNs can learn features to predict selectivity. In Mohamadi et. al. the network needs to explicitly learn what differentiates e.g. zinc from copper binding which is not the case in our work. Within the conceptual framework that we put forward in this work it is therefore clear how one would go about training a model that can predict metal location selectively for each metal (i.e. modify training data, extend the output and add additional loss terms that penalize wrong identity predictions).

We agree that the current lack of selectivity towards zinc is a weakness for certain applications (e.g. protein annotation) but is also not unphysical given the general promiscuity of transition metal ion binding. The lack of selectivity can also be a strength for certain other applications (such

as predictions for rare metals). However, as other reviewers also have raised this point we have modified the title of the paper to put emphasis on the general approach that we put forward in this work as opposed to a selective predictor for zinc or any other specific transition metal: *Metal3D: A general deep learning framework for accurate metal ion location prediction in proteins*.

2. From the paper, it is not clear what Metal1D could be useful for; should it be used in a complementary way to Metal3D? If so, what advantage would it bring to the Metal3D user? Otherwise, I don't see the reason to present two independent tools, one of which is clearly better than the other.

It is common practice for deep learning based models to use a conceptually simple approach as baseline/benchmark. We have therefore chosen to present Metal1D together with Metal3D in this work even though most of the emphasis is on Metal3D due to its better performance and interpretability. Metal1D (<1 sec for prediction on 1 CPU) is much faster than Metal3D (30 seconds per prediction on 1 GPU) and could be used for metal site predictions of extremely large datasets (such as all of ESMAtlas or AlphaFoldDB) to then selectively analyse the hits using Metal3D providing accurate metal ion locations. In addition, Metal1D beats a similar state of the art tool (BioMetAll) giving more precise locations, in faster time with less false positive predictions. We have modified the discussion and conclusion to highlight the different strengths of both tools better.

3. In my experience, the large majority of zinc sites composed by less than three amino acids are largely spurious (at least 60% of cases for sites with two endogenous ligands). The authors should clearly address this point, clarifying whether their tool is intended to identify physiological zinc sites, or any zinc site. In the former case, predicting spurious sites would decrease the precision of the program.

We were initially concerned about the same point that the reviewer raises, especially since we trained on an almost unfiltered collection of metal sites from the PDB. However, in line with the responses to reviewer 1, we think that Metal3D profits from the strength of deep learning in filtering out signal from noisy data in addition to symmetry augmentation of the input environments that ensures that the model sees mostly well organized *in crystallo* metal sites with full coordination and therefore predicts high probabilities >0.75 only for such sites. "In crystallo" sites are not equal to physiological sites however that is why for all analysis we use the biological assembly and report all results split into 'all the zinc sites' (including the ones that have just a single coordination partner because all the other coordinating amino acids are in a symmetry mate in the crystal) and in a subset of sites that resemble physiological sites more closely (2+ or 3+ coordination partners). Figure 3A clearly highlights that most sites that are not found in the complete testset are eliminated when the 2+ criterion is applied and that Metal3D detects 3+ sites with high probabilities ($p > 0.5$) (Figure 3B).

While the model does best on physiological sites with a high degree of preorganization, it can also identify proto-sites with less preorganization (but then with lower confidence). This is especially important for protein engineering applications as the model should be able to score very transient (low probability) to highly preorganized sites (high probability) sites in order to be used for optimization of proteins towards metal binding. We have added concrete recommendations for the cutoff ($p > 0.75$) for identification of physiological sites to the discussion.

4. In the Introduction, the authors state that many tools are available for prediction of metal sites in structures (they list seven different tools, lines 58-60), however they compare their performances with only two tools (MIB and BioMetAll). They should clarify the reason for this selection.

Within the framework of metal site prediction there are multiple different possible tasks. As alluded to in the text (lines 120-124) our tools both predict the xyz coordinates of the metal ions (and a probability density in case of Metal3D) and we therefore only assessed our method against tools

that also provide the coordinates as ultimate output. We reference some other tools that inspired our work but did not benchmark them for the following reasons:

- ZincFinder: does only predict if the protein binds a metal or not and the corresponding webserver is dead <http://zincfinder.dsi.unifi.it/>
- IonCom predicts which residues bind metal but does not provide any location
- AlphaFill is actually not a predictor but simply a homology search + alignment
- FindsiteMetal can predict the location but the server and code are no longer available and not archived anywhere <https://cssb.biology.gatech.edu/findsite-metal/>

5. The authors state that they improved the performance of predictions with respect to existing tools, but do not explicitly report the precision and the recall of their tool, for different p and t values. This would help the user in understanding the results. In this respect, the curve reporting the precision versus recall should be included in the main text of the article, not in the Supplementary Material.

The original reason for not including the precision and recall curve in the main text (apart from the length of the manuscript) is related to the fact that these metrics are not directly computed using the test data but rather within the task of binding site detection. Due to the way how we train Metal3D, the only direct metric of assessing the model is the Jaccard Similarity of the output of the model and the target tensor (Supplemental Figure 1).

We constructed the binding site detection task to compare different methods on the same task - identifying a metal site within a 5 Å radius of a true binding site in a given protein structure. This might potentially be unfair to our methods since the other methods could have possibly seen the structures we test on during training.

As multiple reviewers have raised this point we added the full precision recall curve for all tested tools in the main text together with Figure 3. The numbers change slightly because we also added an occupancy >0.5 criterion for the 2+ and 3+ sets. The curve clearly highlights that Metal3D beats other tools by a good margin (especially on physiological sites (3+)). For MIB vs MIB2, we find a decrease in performance upon extension of the template database in MIB2, most probably due to the loss in precision with only slight gains in recall at low t-values. We still highlight the TP, FP and FN numbers for the highlighted cutoff values since the paper is intended also for non-ML researchers in Figure 3A.

Figure 3B [...] Precision-Recall curve for all tested tools split into all zinc sites in the test set (dotted lines), sites with 2+ unique coordination partners (dashed lines) and sites with 3+ unique coordination partners (solid lines).

6. The Materials and Methods are quite difficult to read, especially for a non-specialized reader. A possible help could be to add a brief description of what the various pre-existing modules used in the program do.

We have updated the Materials and Methods section trying to improve readability and added figures for the individual modules of the two methods hoping that this will make the method section more accessible for non-specialized readers.

Metal3D

Figure 8. Voxelization of an environment: A $C\alpha$ centered box is voxelized using moleculekit library. For each atom an atom centered pair correlation function is used to assign voxel values in 8 different channels (see Table S2)

Figure 9. Sketch of model that predicts residue centered metal densities based on input environment. Number of layers not to scale. Blue tensor sizes indicate number of channels and size of grid ($n_{\text{channels}} \times 32 \times 32$). Convolutional filters (red) are trained and extract information ($3 \times 3 \times 3$ convolutions) or aggregate information ($16 \times 16 \times 16$ convolution)

Figure 10. Grid averaging and metal ion placement: A bounding box for the global grid is defined based on all predictions and residue probability maps are aggregated using KD-Tree search. Ions are placed after AgglomerativeClustering and taking the weighted average of all voxels in a cluster. Probability is the maximum probability of the cluster.

Metal1D

A Coordination environment extraction and probability map generation

B Site prediction

Figure 7. Metal1D method A Coordination environment extraction from LINK records of PDB files and probability map generation **B** Site prediction, showing examples for a Zn²⁺ binding site, correctly predicted, and unlikely Zn²⁺ binding site, discarded in the putative site scoring step because of low probability of the metal site, and an amino acid for which Zn²⁺ binding is impossible, discarded in the residue scoring step because of low score of the amino acid.

Reviewer #3 (Remarks to the Author):

This manuscript presents the development of a new deep learning-based tool for the prediction of metal binding sites. It appears to have been thoroughly developed and addresses a difficult problem in an original way. I have the following comments:

1. As mentioned in the manuscript, metals are often present in the buffers used for crystallisation. What steps have been taken to ensure that the metal binding sites used in training and testing are biologically relevant?

We were also concerned about this issue which is why we discuss data quality in the discussion of our manuscript. Since reviewer 1 and reviewer 2 have also raised similar concerns about data quality, we have extended this discussion and have also provided some problematic examples in Supplemental Figure S7. Specifically relating to crystal artefacts from crystal buffers, we use the symmetry related structures to restore the metal sites in the training to the fully coordinated "in crystallo" environment. When the model is presented with a biological assembly that contains artefact sites on the surface with coordination partners from symmetry mates, it does predict them only with low probability (Supplemental Figure S7). During evaluation we show results with such sites included/excluded ('all Zn sites' vs '2+' and '3+' categories). We now provide guidance in the discussion, which probability cutoff ($p > 0.75$) and which structure (the biological assembly) should be used to predict biologically relevant sites.

2. I think the results would benefit from a brief explanation of the dataset i.e. how many binding sites are present in the test set and from how many different proteins?

We thank the reviewer for this helpful suggestion. We now provide statistics on the training and test set with respect to resolution, occupancy of the zinc ions, multi-nuclear zinc sites, number of zincs in the structure and other metals (Figure S9-S19). The number of proteins in the test and train set was already contained in the text (1,93, 1,145f) and the concrete PDB identifiers are provided in the supplementary material in a csv file. Since we use the full PDB clustered at 30 % sequence identity we train on the full diversity of zinc proteins. The 30 % identity cutoff was

chosen since it eliminates bias towards certain protein families that have been structurally characterized much more often than others (e.g antibodies).

3. On the dataset - reading the methods I get the impression that testing/validation was performed on a very small number of protein structures - clearly a large number of structures are required for training but to have only 85 proteins for testing, which are further split seems very small. Is the set that performance with other methods is compared? If so, then this small dataset could be biased and not represent performance of any of the methods on a larger dataset.

We agree that the test dataset is on the smaller side for deep learning projects, however the number of different coordination motifs observed for zinc in the training set with more than 50 examples is only 62. Our testset which only contains unrelated protein families with not even partial sequence overlap to the training set should therefore contain the majority of them (189 metal sites in 59 proteins).

In addition some of the methods we test against (MIB, MIB2) required us to manually upload each structure to a server so it was impractical to have a larger test set. We now provide explicit statistics of the distribution of coordination motifs in the test and training set in Supplemental Figure S19 showing that the two most common zinc motifs (CCCC and CCCH) have similar distribution in train and test. Most of the other less common motifs (<5% frequency) also have at least 1 example in the test set. Therefore, although small, our test set should contain a realistic sample of zinc sites from proteins that do not even have partial sequence overlap with the proteins we train on.

Figure S19. Distribution of the coordination motifs for the biological assemblies included in the training set (A1) and the test set (A2) sorted by frequency and overlaid sorted by frequency of the coordination motifs in the training set (B).

4. Greater clarity is needed on the presentation of the performance of the Metal1D and Metal3D, especially compared to other methods. Figure 3 is difficult to interpret and also makes extensive use of the combination of red and green. The text and this figure also refer to t , c and p as values but it is not clear what these are, this needs to be explained. The only one that appears to make sense is p , which appears to relate to probability. Given that these deep learning methods are likely to associate each prediction with a probability or confidence score, I would think it would be much easier to understand to present a precision-recall graph (present in supplementary figure 2) of the results rather than the difficult to interpret bar chart. Then looking at the precision-recall graph the performance is not that good - at a high probability threshold a recall of 40% is obtained with precision close to 1 (this is fairly good) but at the lowest threshold - recall only reaches 60% at precision of 40%.

A similar point has been raised by Reviewer 2. We have improved clarity of the text on the binding site prediction task which is a posterior task to compare the different methodologies with each other. Neither Metal3D nor Metal1D is directly a classifier and we cannot directly compute the precision recall curve using the test data wherefore we had originally chosen to report the precision recall curve only in the supplementary material as it is helpful to determine the probability cutoff to use. However, since multiple reviewers have raised this point, we have now included the full precision recall curve for the binary binding site prediction task in the main text in Figure 3B. Each curve was obtained by varying the determining parameter for each method (see Table S5 for an explanation). The precision recall curve now includes results for the different categories in the test set (all, 2+ and 3+). Compared to the state of the art predictors BioMetAll and MIB2, Metal3D is significantly better. On the 3+ subset (all structures in test that contain at least 1 physiological site) at the recommended cutoff for physiological sites ($p=0.75$), Metal3D achieves precision 0.81 and recall 0.79.

Figure 3B [...] Precision-Recall curve for all tested tools for the binding site task split into all zinc sites in the test set (dotted lines), sites with 2+ unique coordination partners (dashed lines) and sites with 3+ unique coordination partners (solid lines). Determining parameters for each method are described in Table S5.

Table S5. Main parameter for each method. Arrows indicate in which direction higher quality results are obtained (lower recall but higher precision).

Method	parameter	description
Metal3D	Probability $p \uparrow$	Probability value predicted by the Metal3D model
Metal1D	Threshold $t \downarrow$	Threshold that is used twice for the scoring to remove residues in step 1 (residue scoring) that have score lower than $(1-t) \times \text{max residue score}$ and in step 2 (putative site scoring) to remove sites that have score lower than $(1-t) \times \text{max site score}$
BioMetAll	Cluster cutoff $c \uparrow$	Denotes the number of probes predicted for a given cluster divided by the number of probes for the highest scoring cluster
MIB/MIB2	Template similarity $t \uparrow$	Measure of sequence and structure conservation of the template site in reference to the predicted site

5. Binding sites predicted with 5 angstroms of an actual binding site are classed as true positives - why was this distance selected, is it too lenient? What is the effect of reducing this threshold?

We chose this distance as it is about the double of an average zinc coordination distance of $\sim 2.2 \text{ \AA}$. It allows for fair comparison of deep learning, distance and template based predictors. We agree that it is important to also show how accurate the predicted position in comparison to the experimental position is (not just in the binary case within 5 \AA or not). Figure 4A shows the distribution of the distance deviation of the correctly predicted locations to the exact experimental location. Some tools like MIB predict many different locations and we show the deviation of each predicted position in the distribution and indicate the number of predicted locations to the right of each distribution ("n="). We have made this clearer in the caption of Figure 4A. From Figure 4A it is evident that for Metal3D and MIB the distance threshold could be reduced to about 1.5 \AA without changing the results since most predictions are below this threshold.

Figure 4. Mean absolute deviation of correctly predicted sites and selectivity for other ions

A For all sites where predicted and experimental location are available (true positives in Figure 3) we compute the mean absolute deviation (MAD) using Metal1D, Metal3D, BioMetAll, MIB and MIB2 on the test set. Some tools predict multiple locations (n) per site (TP). *For MIB and MIB2 we used 2 structures less because the server did not accept these structures. For each tool the whisker plot indicates the median (white dot) and the first quartiles (black box). [...]

6. What is the purpose of the section on selectivity for other metals? This needs to be clearer. Why were only 25 structure selected - this is a small number of structures to test on, there must be many more available? . Figure 4B shows the recall for these biding sites but this is not very informative without precision as well. Again the manuscript should be presenting the performance in a way that is easy to interpret.

We train only on zinc and therefore deemed it necessary to check how selective our tool is for other metals because we only enforced selectivity implicitly. Users of Metal3D might naively expect that it only detects zinc sites since we train on zinc only.

The model should learn selectivity for zinc since we train it by sampling half of the environments from non-zinc binding residues of the training proteins. These non-zinc environments included residues that bind other metals and the other metal was not explicitly modelled. The selectivity analysis was necessary to show that this implicit training for zinc selectivity is not enough to get selective predictions for zinc because the coordination environments that can support zinc binding are too similar to the environments that support binding of other transition metal ions (in line with the Irving Williams Series and how metal selectivity is enforced in nature e.g ref1, ref2, ref3). Recent work by Mohamadi et.al (ref4) shows that using the same input a 3DCNN can indeed learn selectivity, so the framework we put forth in this work could easily be extended to predict a $n_{\text{metal}} \times 32 \times 32 \times 32$ tensor and training it with other metal ions. Other reviewers also have raised this point and we updated the discussion and title to make this clearer.

We agree that 25 structures might be too small a sample and we therefore extended the selectivity dataset to up to 100 biological assemblies (sampled from the train/test split in the 30% clustered pdb). For NI (93), CU(68), FE2(57) and CO(30) less structures were used because those were the only ones fulfilling the criteria (minimum one site with 3 distinct protein residues coordinating the ion $<2.8 \text{ \AA}$ (for transition metals and earth-alkali) and $<4\text{\AA}$ for alkali ions). The updated cutoff for alkali ions was necessary to find enough structures because alkali ions typically have larger coordination distances.

We thank the reviewer for suggesting to include the precision information in Figure 4B since it is indeed informative. We modified Figure 4B to include the updated set of structures, included the occupancy cutoff (>0.5) for all sites and changed the probability cutoff for Metal3D from 0.5 to 0.75 (for physiological sites) for the selectivity analysis. From the new figure it is evident that Zn^{2+} is the metal where both methods have the highest precision and recall with Metal1D featuring a more significant gap between Zn^{2+} and the other transition metals. The original conclusion remains unchanged that the way we trained Metal3D is not enough to enforce high selectivity towards zinc and the model identifies most well defined transition metal ion binding sites. We are currently working on the extension to selectively predict metal location

Figure 4B: [...]Precision and recall for the test set and 100 randomly drawn structures (93 NI, 68 CU, 57 FE2, CO 30) for other transition (blue), earth-alkali (pink) and alkali (brown) metals. Metal1D as light squares, Metal3D as circles. Threshold used for Metal3D $p = 0.75$, threshold used for Metal1D $t = 0.75$

REVIEWERS' COMMENTS

Reviewer #1 (Remarks to the Author):

After carefully checking the response prepared by the authors, I think all my concerns have been well addressed. Thus I recommend the acceptance of this article in Nature Communications.

Reviewer #2 (Remarks to the Author):

The authors answered all my concerns and the work now appears well constructed and well discussed. Although the reported performances are not excellent, it is certainly a step forward in the prediction of zinc sites in structures and so I believe it is appropriate for your journal

Reviewer #3 (Remarks to the Author):

The revised manuscript has addressed all of the comments that I have raised.

Signed - Mark Wass